# Uncovering hidden network architecture from spiking activities using an exact statistical input-output relation of neurons

Safura Rashid Shomali [1✉], Seyyed Nader Rasuli[2,3], Majid Nili Ahmadabadi[4] & Hideaki Shimazaki [5,6✉]

Identifying network architecture from observed neural activities is crucial in neuroscience studies. A key requirement is knowledge of the statistical input-output relation of single neurons in vivo. By utilizing an exact analytical solution of the spike-timing for leaky integrate-and-fire neurons under noisy inputs balanced near the threshold, we construct a framework that links synaptic type, strength, and spiking nonlinearity with the statistics of neuronal population activity. The framework explains structured pairwise and higher-order interactions of neurons receiving common inputs under different architectures. We compared the theoretical predictions with the activity of monkey and mouse V1 neurons and found that excitatory inputs given to pairs explained the observed sparse activity characterized by strong negative triple-wise interactions, thereby ruling out the alternative explanation by shared inhibition. Moreover, we showed that the strong interactions are a signature of excitatory rather than inhibitory inputs whenever the spontaneous rate is low. We present a guide map of neural interactions that help researchers to specify the hidden neuronal motifs underlying observed interactions found in empirical data.

[1] School of Cognitive Sciences, Institute for Research in Fundamental Sciences (IPM), Tehran 19395-5746, Iran. [2] School of Physics, Institute for Research in Fundamental Sciences (IPM), Tehran 19395-5531, Iran. [3] Department of Physics, University of Guilan, Rasht 41335-1914, Iran. [4] Control and Intelligent Processing Center of Excellence, School of Electrical and Computer Engineering, College of Engineering, University of Tehran, Tehran 14395-515, Iran. [5] Graduate School of Informatics, Kyoto University, Kyoto 606-8501, Japan. [6] Center for Human Nature, Artificial Intelligence, and Neuroscience (CHAIN), Hokkaido University, Hokkaido 060-0812, Japan. ✉email: safura@ipm.ir; h.shimazaki@i.kyoto-u.ac.jp

One goal of neuroscience is to expose in vivo neural circuitries by using recorded neuronal activities. The recent technological advances in connectome projects have revealed complete wiring diagrams of model animals[1,2]. Nonetheless, to determine what computations certain neural circuitry performs in living systems, it is still important to identify the network architecture from in vivo recordings of multiple neurons[3]. Simultaneous intracellular recordings are the most reliable way to identify physical connections in vivo[4–8]. Here, one should simultaneously record neurons and all their presynaptic inputs by using the patch-clamp technique to find influential synapses. However, as this technique can only be applied to a very small subset of neurons, it can identify a few connections. An alternative approach to find connections is to use extracellular recordings and imaging methods to acquire simultaneous neuronal spiking activities of a large number of neurons[9]. Cross-correlograms[10,11] or constructing point-process network models are classical ways of inferring connectivity from spiking data[12–14]. However, these methods aim to discover connections among the recorded neurons, whereas the majority of synaptic inputs come from unobserved neurons. Therefore, it remains a challenge to reveal the hidden neuronal circuitries by using the activity statistics of a limited number of neurons in vivo.

The hallmark of cortical spiking activity in vivo is its variability[15,16]. It has been suggested that the variability of spiking activity is the result of balanced inputs from excitatory and inhibitory neurons fluctuating near the spiking threshold[16–18]. Such balanced inputs have been confirmed by intracellular recordings of the sensory-evoked activities of in vivo neurons in rats[19] and monkeys[20]. Under such conditions, even a moderate synaptic input can cause a spike in the postsynaptic neuron. However, as the distribution of the synaptic strengths in cortical and hippocampal neurons follows a log-normal distribution[21], it was suggested that fewer strong synaptic connections constitute the backbone of the microcircuit, with the aid of inputs from a large number of weak synapses[22–26].

Given this common picture of cortical variability, we need to discover the architecture of the influential synapses in order to reveal the basic motifs of microcircuits operating in vivo. The current models that link architecture to the statistics of neural activity assume weak synapses and linear responses to the synaptic input[27–33] (but see refs. [34,35]). However, the nonlinearity of input-output relation means that we cannot use linear-response methods to identify the influential inputs. Here, we can instead use the recent analytical solution for the leaky integrate-and-fire (LIF) neuron model that includes the dependency of output spikes on arbitrary synaptic inputs of interest, whereas the effects of many weak synapses accumulate as noisy background inputs, balancing neuron's voltage near the spiking threshold. It predicts that a strong synaptic input results in a nontrivial response different from the weak/moderate inputs[36].

To reveal the hidden neuronal motifs, we need a framework for judging how the hidden network of input neurons shapes the complex joint activity of postsynaptic neurons, possibly characterized by their higher-order interactions[37,38]. This framework, in turn, could be used as a tool to infer the hidden architecture from population statistics of observed postsynaptic neurons. Here we aim to gain insight into the underlying architecture from higher-order neuronal interactions, i.e., interactions among three or more neurons, because occurrence of significant higher-order interactions have been found ubiquitously in vitro[39–41] and in vivo[42–45], and predicted by theoretical studies[46–50]. Furthermore, it has been reported that the higher-order interactions encode stimulus information[43,51] (see also refs. [52,53] for a simulation study) and relate to animals cognitive functions such as expectation[54], perceptual accuracy[55], and prediction[56]. Thus, they potentially provide important clues on the architecture of cortical circuitries functioning in living systems.

In this study, we used the aforementioned analytic solution[36] to construct a framework of network identification from observed pairwise and higher-order interactions in spiking activities of neurons. We looked at the simplest scenario and tried to answer the following questions: the experimentalist records spiking activities of three neurons in vivo (e.g., ref. [43]) while s/he cannot directly reveal any synaptic connectivity. Do the three neurons spike independently or show correlations due to possible shared inputs? In the latter case, are such inputs shared between each individual pair or among all three? Are the shared inputs excitatory or inhibitory? And finally, does any of the three observed neurons make a direct synaptic connection to another of trio? Using the biophysical LIF model, we show that it is possible to determine the architectures of hidden shared inputs by carefully examining pairwise and triple-wise interactions of the three neurons. Moreover, we determine model-free boundaries that each architecture occupies in the space of neuronal interactions, with which one can unambiguously identify the underlying motif, if the interactions are significant. The predicted analytical regions were validated using the blue brain multicompartment neuron model[3,57].

We compared the theoretical predictions with experimentally observed neuronal interactions in the V1 areas of monkeys and mice. Here, Ohiorhenuan et al. found significant positive pairwise and negative triple-wise interactions for spatially close neurons in V1 area of monkeys[43,58], and our analysis of awake mouse V1 neurons[59] showed similar results. The negative triple-wise interactions, observed in cortical and hippocampal neurons[41–43], indicate a significantly higher probability of simultaneous silence among the three neurons than would be expected from their rates and pairwise correlations. Intuitively, common inhibitory inputs should induce an excess of simultaneous silence by suppressing neurons. However, by superimposing the data on the plane of pairwise and triple-wise interactions with analytic boundaries for motifs, we quantitatively ruled out shared inhibition as the motif underlying the observed strong negative triple-wise interactions. Rather, the data supported a non-intuitive architecture of common excitatory inputs, shared by pairs of neurons (excitatory inputs to pairs). We investigated how our conclusions are affected by the presence of directional/recurrent connections among observed neurons, and by considering adaptive neurons. We confirmed that many of these results, particularly the significance of the motif of excitatory inputs to pairs, remain valid.

Overall, our framework can be used as a quantitative tool to reveal hidden neuronal microcircuits. In particular, we have summarized all of results into a unified guide map in which each motif occupies its own region in the space of neuronal interactions. This guide map will help experimentalists identify hidden motifs underlying the correlated neuronal activities observed in their experiments.

## Results

**Spike density of in vivo LIF neurons for modeling population activity driven by common inputs**. A shared input to two postsynaptic neurons marks its presence in their correlated activity, and neurophysiologists often record the activity pattern of pairs of neurons, in the hope of determining the existence of possible shared input, the input's type and strength. This requires a mathematical framework to answer the questions (**a**) how a presynaptic input, weak or strong, modifies the activity of a postsynaptic neuron, and (**b**) how correlated activity among postsynaptic neurons emerges when they do share such an input.

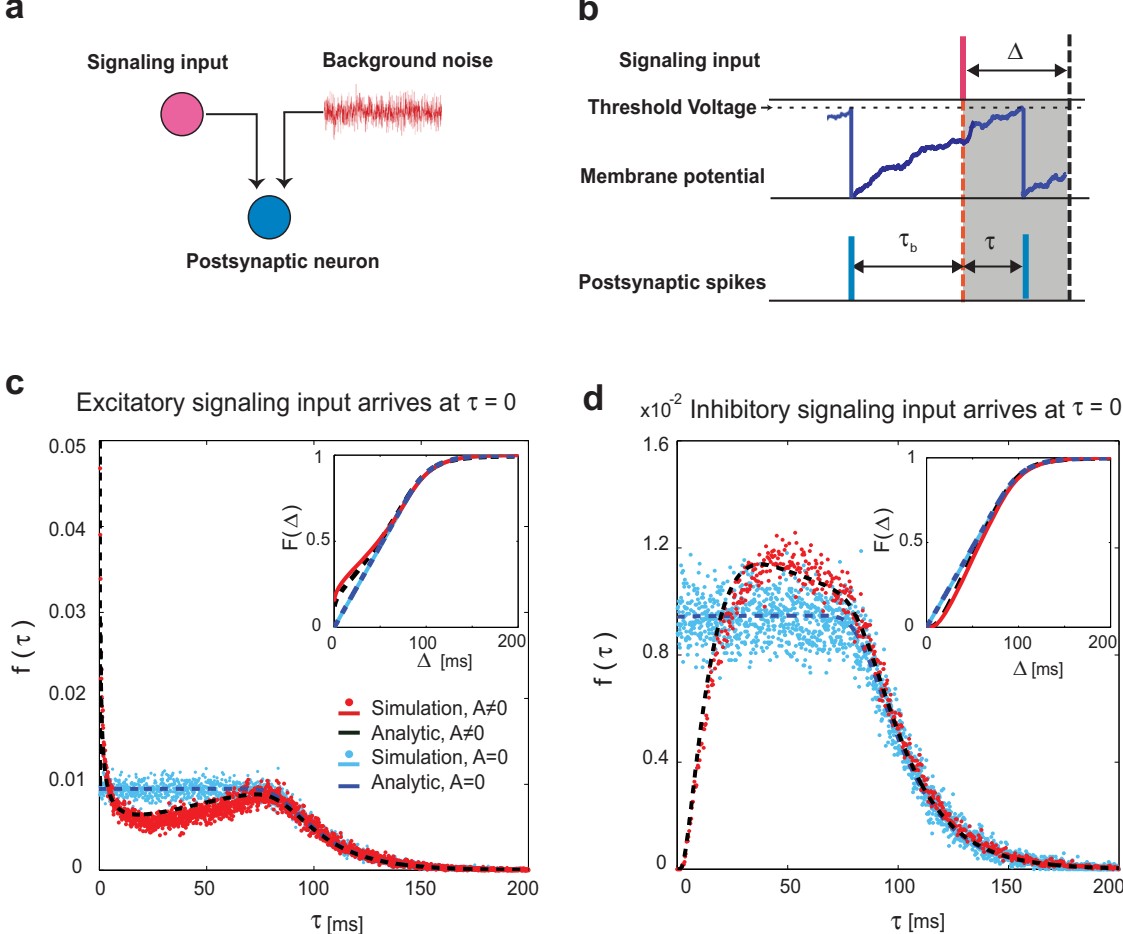

**Fig. 1 Analysis of a postsynaptic neuron receiving signaling input on top of background noise. a** Schematic model of postsynaptic neuron driven by background Gaussian noise and signaling input. The postsynaptic neuron has a threshold potential of $V_\theta = 20$ mV, membrane time constant $\tau_m = 20$ ms, and it receives a signal of amplitude $A = 5$ mVms; the background input fluctuations are quantified as $D = 0.74$ mV$^2$ms (Methods). The values of the parameters are chosen from physiologically plausible ranges[106]. **b** Timing of postsynaptic spikes and arrival time of signaling input. The postsynaptic neuron generates a spike (blue tick); then, its membrane potential resets. While the potential is rising, a signaling input (red tick) arrives at $\tau_b$ after the last spike, which changes the trajectory of the membrane potential. The postsynaptic spike occurs at time $\tau$ after the arrival of the signaling input. The gray shaded area of width $\Delta$ indicates the observation time window during which the probability of postsynaptic activity pattern is computed. **c, d** Probability densities of the first spike occurring at $\tau$ after arrival (Eq. (12), Methods) of excitatory (**c**) and inhibitory (**d**) signaling input. The simulation results illustrated as red dots, and the analytical solutions (Eq. (12)) are shown as dashed black lines. The densities for a signaling input with zero amplitude are plotted as light blue dots (simulation) and dashed blue lines (analytical solution). Inset: Probability of spiking within $\Delta$ after signaling input arrival (Eq. (13)).

Here, to predict how a presynaptic input affects the activity of postsynaptic neurons (the question **a**), we devised a framework for computing the statistical properties of the activity of LIF neurons (Methods). Then, to see how correlated activity emerges (the question **b**), we studied interactions between two (in this section) and among three postsynaptic neurons (in the next section). In particular, we investigated how a common excitatory or inhibitory input with an arbitrary strength on top of independent noisy background inputs causes the spiking activities of the postsynaptic neurons to be correlated.

Figure 1a shows a schematic image of the in vivo neuron model we used. The neuron receives signaling inputs with arbitrary efficacy (strength), on top of noise composed of many weak synaptic inputs that brings the neuron's equilibrium membrane potential close to the threshold. Each postsynaptic neuron is modeled using the LIF model with a threshold potential of $V_\theta$ and membrane time constant $\tau_m$ (Methods: Effect of presynaptic spike-timing on leaky integrate-and-fire neuron receiving noisy inputs balanced near threshold). The noisy background inputs are

approximated by a Gaussian distribution with a mean drive of $\bar{I}$ and the variance of $2D/\tau_m$, where $D$ [mV$^2$ms] is the diffusion coefficient. In addition, the neuron receives a transient signaling input of arbitrary amplitude $A$. Figure 1b illustrates the time course of membrane potential, confronted with signal arrival time and postsynaptic neuron's spike time. We split the question of an individual neuron's response, i.e., the question **a**, into two parts: first, what is the probability density of a spike occurrence at time $\tau$ after signal arrival, $f_A(\tau)$? And second, what is $F_A(\Delta)$, the probability of observing one or more spikes in a time window of $\Delta$, after the occurrence of presynaptic signal with strength $A$? Here, $F_A(\Delta)$ is a quantity that predicts the observed spiking activity of neurons from the model. Using $F_A(\Delta)$, one can find the probability of various spiking patterns for multiple neurons receiving common signaling inputs. We used the solution of the spiking density[36] (Methods: Spiking density of leaky integrate-and-fire neuron receiving signaling input in the threshold regime) to calculate $F_A(\Delta)$ analytically (Methods: Spiking density of LIF neuron after signaling input arrival).

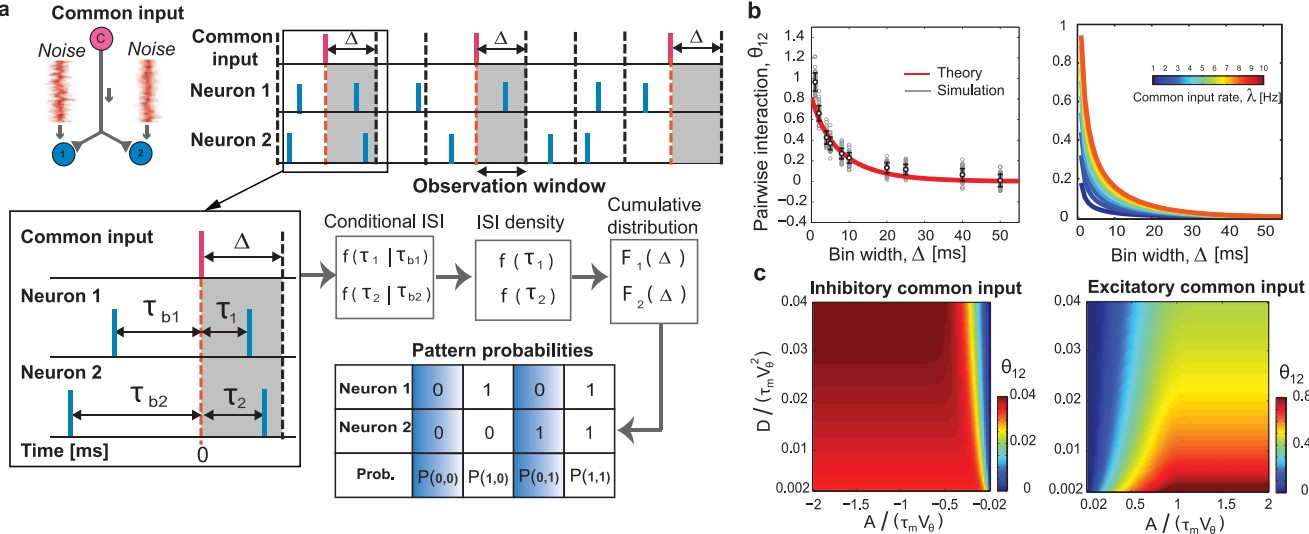

**Fig. 2 Analysis of pairwise interaction of two neurons receiving common signaling input on top of background noise. a** Top, left: Schematic model of two postsynaptic neurons (Neuron 1 and Neuron 2, blue circles) driven by independent noise and common signaling input (pink circle). Top, right: Spike trains of two postsynaptic neurons (blue spikes) receiving common signaling inputs (red spikes) with rate $\lambda$. The gray shaded area of width $\Delta$ indicates the time window during which the probabilities of the postsynaptic activity patterns after the signaling input are computed. Bottom, left: Timing of postsynaptic spikes relative to the arrival time of the common input. The last spike of Neuron 1 (Neuron 2) occurs $\tau_{b1}$ ($\tau_{b2}$) before the arrival of the common input, and the next spike happens at $\tau_1$ ($\tau_2$). Middle: Conditional spike density after input arrival is calculated using Eq. (11). By marginalizing over the previous spike ($\tau_b$), one obtains the probability of spiking after arrival of the input (spike density, Eq. (12)). The next step is to calculate the cumulative distribution function (Eq. (13)), which is the probability of having one or more spikes within the time window $\Delta$. Bottom: Probability of having a particular pattern of spikes for two neurons is obtained, based on the cumulative distribution function and the fact that neurons are conditionally independent. Four possible binary activity patterns (00, 01, 10, 11) of two postsynaptic neurons and their associated probabilities ($P(i,j)$, $i,j \in \{0,1\}$). '1' denotes the occurrence of at least one spike within the time window $\Delta$, whereas '0' represents silence of the neuron within this window. **b** Pairwise interaction ($\theta_{12}$, Eq. (15)) as a function of time window, $\Delta$. Here we used a physiologically plausible range of parameters, $V_\theta = 20$ mV, $\tau_m = 20$ ms, $A = 5$ mVms, and diffusion coefficient $D = 0.74$ (mV)$^2$ms[106]. Left: Pairwise interaction computed from simulated spike sequences (gray lines: 50 trials each containing about 2500 spike occurrences of common input; dots and error bars: mean ± standard deviation) compared with the analytic result of the mixture model (red line, Eq. (15)) for common input rate $\lambda = 5$ Hz. Right: Analytical value of $\theta_{12}$ as a function of bin size, $\Delta$ for different common input rates, $\lambda$. **c** Pairwise interaction as a function of the scaled diffusion coefficient, $D/(\tau_m V_\theta^2)$, and shared signal strength, $A/(\tau_m V_\theta)$, for excitatory (right) and inhibitory (left) common inputs with rate, $\lambda = 5$ Hz and bin size, $\Delta = 5$ ms.

The dashed black curves in Fig. 1c (or d) show the spiking density at time $\tau$ after signaling input arrival, $f_A(\tau)$, for square-shaped inputs of the excitatory (or inhibitory) type (Eq. (10) in Methods). Compared with the no signaling input case (analytically), it is more (or less) probable that a spike will occur at small $\tau$ after arrival of an excitatory (or inhibitory) input. At sufficiently large $\tau$, however, the spiking densities with and without the signaling input are virtually identical, indicating the brief effect of the signaling input. Accordingly, the cumulative distribution functions, $F_A(\Delta)$, (Fig. 1c and d insets) with and without the signaling input differ for small $\Delta$, but are indistinguishable for large $\Delta$. This result implies that we cannot discern the presence of a signaling input if we use a large time window. Note that these analytic results were confirmed by numerical simulation of the LIF equation (blue and red symbols).

With the above analytical knowledge about $F_A(\Delta)$ verified by the simulation, we investigated the question **b**: emergence of correlations among neurons receiving shared input. In this section, we first consider two postsynaptic neurons for simplicity; in the next section we extend the framework to three neurons. Suppose that the two postsynaptic neurons receive a random common signaling input, in addition to independent background noise. We assume that the firing rate of the common input $\lambda$ is not large so that we can safely consider it to be a sparse input. Figure 2a illustrates the timing of the postsynaptic spikes before and after the common input arrival. We segment the spike sequences by using time window of size $\Delta$. For simplicity, we assume that the onsets of the common input are aligned at the

bins. To label various spiking patterns, we use a binary variable $x_i \in \{0, 1\}$, where $i = 1, 2$ indexes the two postsynaptic neurons. $x_i = 1$ means that the $i$th neuron emitted one or more spikes in the bin, while $x_i = 0$ means that it remained silent. Accordingly, $P_A(x_1, x_2)$ measures the probability of the spiking pattern, defined by $x_1$ and $x_2$, due to onset of a common input of strength $A$. For instance, $P_A(1, 0)$ is the probability that one postsynaptic neuron has spiked and the second remained silent, in the time window $\Delta$ after onset of a common input of strength $A$.

Apart from the common input, the rest of the two postsynaptic neurons' inputs are independent; i.e., they are driven independently by background noisy inputs. Thus, the probability of an activity pattern occurring is given by:

$$P_A(x_1, x_2) = P_A(x_1) \times P_A(x_2) = \prod_{i=1}^{2} F_A(\Delta)^{x_i} (1 - F_A(\Delta))^{1-x_i}. \quad (1)$$

Note that the signal amplitude $A$ is the only common factor between $P_A(x_1)$ and $P_A(x_2)$. The combination of $F_A(\Delta)^{x_i}(1 - F_A(\Delta))^{1-x_i}$ reduces to $F_A(\Delta)$ when $x_i = 1$, i.e., the $i$th neuron has spiked, and $1 - F_A(\Delta)$ when the neuron is silent. The rate of the common input ($\lambda$) is applied to $\lambda \Delta \times 100\%$ of the bins whereas it is absent in $(1 - \lambda \Delta) \times 100\%$ of the bins. Here we assumed sparse signaling inputs so that the probability of more than one signaling input in a single bin is small.

We modeled the spike sequences of two postsynaptic neurons as a mixture of two situations: either two neurons receive common input ($A \neq 0$), or they do not receive it ($A = 0$).

Consequently, the probability is a combination of two conditions with weights given by the occurrence probability for each situation:

$$P(x_1, x_2) = \lambda\Delta\, P_A(x_1, x_2) + (1 - \lambda\Delta)\, P_0(x_1, x_2) \qquad (2)$$

Note that this mixture model gives approximate probabilities of the activity patterns of the LIF neurons because, if a neuron does not spike within $\Delta$ [ms] after the common input, the effect of the augmented/reduced membrane potential is carried over to the next bin, and thus the binary activities are no longer a simple mixture of the two conditions (Supplementary Fig. 1 and Supplementary Note 1). Such situations often happen if the bin size is small compared with the mean postsynaptic inter-spike interval.

The strength of interaction can be determined by writing the probability distribution in the form of the exponential distribution[48,60]:

$$P(x_1, x_2) = \exp(\theta_1 x_1 + \theta_2 x_2 + \theta_{12} x_1 x_2 - \psi), \qquad (3)$$

where $\theta_1$ and $\theta_2$ are the individual parameters of two neurons, $\psi$ is a normalization factor, and $\theta_{12}$ is the pairwise interaction. For two neurons, the probabilities of the activity patterns can be constructed from the probability of spiking in a time window $\Delta$ (Eq. (13), Fig. 2). From this probability mass function, Eq. (2), one can compute the neuron's pairwise interaction, denoted as $\theta_{12}$ (Eq. (15), Methods: Pairwise and triple-wise interactions of neural populations). To investigate the neuronal correlation, we used this information-geometric measure of the pairwise interaction[37,61–63] (Eq. (15)). We selected this measure because it is not correlated with the estimated firing rates, whereas the classical covariance and correlation coefficient estimations are (i.e., the pairwise interaction in this method is orthogonal to firing rates in terms of the Fisher metric; see Supplementary Note 2).

We checked if the approximate mixture model predicts the interaction in the sequences of the two LIF neurons and determined appropriate bin sizes. Figure 2b compares the pairwise interaction predicted by the mixture model with the simulated spike sequences. It displays the interaction for different bin sizes when the two neurons receive common excitatory input. The pairwise interactions predicted by the mixture model are within the error bars of the simulation except for the smallest bin (1 ms, left panel, red and gray lines, respectively). The result also shows $\theta_{12}$ increases with the rate of the common input (Fig. 2b, Right). However, the probability of having one or more spikes within $\Delta$ increases for larger bin sizes and saturates to 1 (Fig. 1b, Inset) regardless of the presence or absence of the signaling input. Thus, $F_A(\Delta)/F_0(\Delta) \to 1$, which means that the pairwise interactions vanish and we cannot use the binary representation to determine whether there is a common input when the bin size is large.

We further examined the pairwise interactions by changing two independent parameters, the scaled amplitude of the signaling input $A/(\tau_m V_\theta)$ and the scaled variability of the noisy background input $D/(\tau_m V_\theta^2)$ (Fig. 2c). As expected, the pairwise interactions were positive for both common excitatory and inhibitory inputs. However, the interactions were significantly weaker in the inhibitory case. This indicates that it is difficult to observe the effect of a common inhibitory input for this range of postsynaptic firing rates and that strong pairwise interactions are indicator of common excitatory inputs. We verified this trend by simulating two LIF model neurons receiving a shared Poisson input on top of noisy inputs that balanced the voltage near the threshold regime (Supplementary Fig. 2a, d, see also Supplementary Note 1).

Moreover, as shown in Fig. 2c, there exists a critical normalized amplitude for common excitatory input, $A/(\tau_m V_\theta) \sim 1$ for each

value of scaled diffusion coefficient (level of inputs' noise), $D/(\tau_m V_\theta^2)$. Above this critical value, the postsynaptic neuron's spiking density, and consequently the pairwise interaction, does not change anymore (Fig. 2c, right). The saturation value of the pairwise interaction is inversely correlated with the scaled diffusion coefficient: since a higher scaled diffusion coefficient (noise level) disperses the membrane voltage, the probability of spiking decreases after the common input arrival. In contrast to the common excitatory input, where the effect of the common input is stronger for a low scaled diffusion coefficient (low firing rate), we can see the effect of the common inhibitory input is stronger for higher scaled diffusion coefficient (Fig. 2c, left). This observation is discussed in the section titled "Excitation versus inhibition: which one can produce stronger triple-wise interactions?"

**Higher-order interactions among three neurons depend on type of common inputs and network architecture.** Here, we extend the above analysis of neural interactions to three neurons. Our motivation to investigate the interactions among three neurons comes from the results of experimental studies[43,58] that investigated the simultaneous activities of three neurons (Fig. 3a). It was shown that for two neurons, there is one possible shared input architecture: a common input to both; whereas for three neurons, it can be either (i) a shared input among the three (red connections in Fig. 3a), or (ii) one or more shared inputs to each pair among them (green connections). Assuming symmetry, the former is the star architecture or excitatory (or inhibitory)-to-trio (Fig. 3b, left) while the latter is the triangle architecture or excitatory (or inhibitory)-to-pairs (Fig. 3b, right).

To quantify the neuronal correlation among three neurons, we investigated the information-geometric measure of the triple-wise interaction, $\theta_{123}$, by using the log-linear model[37,61,63] (Eq. (17)). Similar to the pairwise interaction, estimation of the triple-wise interaction measure is not affected by (i.e., it is orthogonal to) the estimated individual firing rates or joint firing rates of two neurons, and therefore it reveals the pure triple-wise effect that cannot be inferred from the first and second-order statistics of the neuronal population (Supplementary Note 2). There are two basic motifs that can induce triple-wise interactions among the three neurons (Fig. 3b, left and right), as described below.

*Common input to three neurons: Star architecture.* Three neurons simultaneously receive a single common signaling input (Fig. 3c). The conditional probability of the activity patterns when a common input generates a spike given to all three neurons with probability $\lambda\Delta$ (red lines) is $P_A(\mathbf{x}) = \prod_{i=1}^{3} F_A(\Delta)^{x_i}(1 - F_A(\Delta))^{1-x_i}$, where $\mathbf{x} = (x_1, x_2, x_3)$ is the spiking activity of three neurons. Similarly, the probability mass function for three neurons receiving no common input with probability $1 - \lambda\Delta$ (gray dashed lines), is $P_0(\mathbf{x}) = \prod_{i=1}^{3} F_0(\Delta)^{x_i}(1 - F_0(\Delta))^{1-x_i}$. Thus, we model the spike occurrence as a mixture of the two conditions in which neurons receive and do not receive common input:

$$P(\mathbf{x}) = \lambda\Delta P_A(\mathbf{x}) + (1 - \lambda\Delta)P_0(\mathbf{x}). \qquad (4)$$

From this probability mass function, we can compute the triple-wise interaction of three neurons according to Eq. (17) (Methods: Pairwise and triple-wise interactions of neural populations).

*Common inputs to pairs of three neurons: Triangle architecture.* Each pair of neurons among trio receives a common signaling input from an independent presynaptic neuron with frequency $\lambda$ (Fig. 3d). Neurons 1 and 2 share one input in common, as do neurons 2 and 3 and neurons 1 and 3 (symmetric case). The three common inputs are independent and occur with equal frequency, $\lambda$.

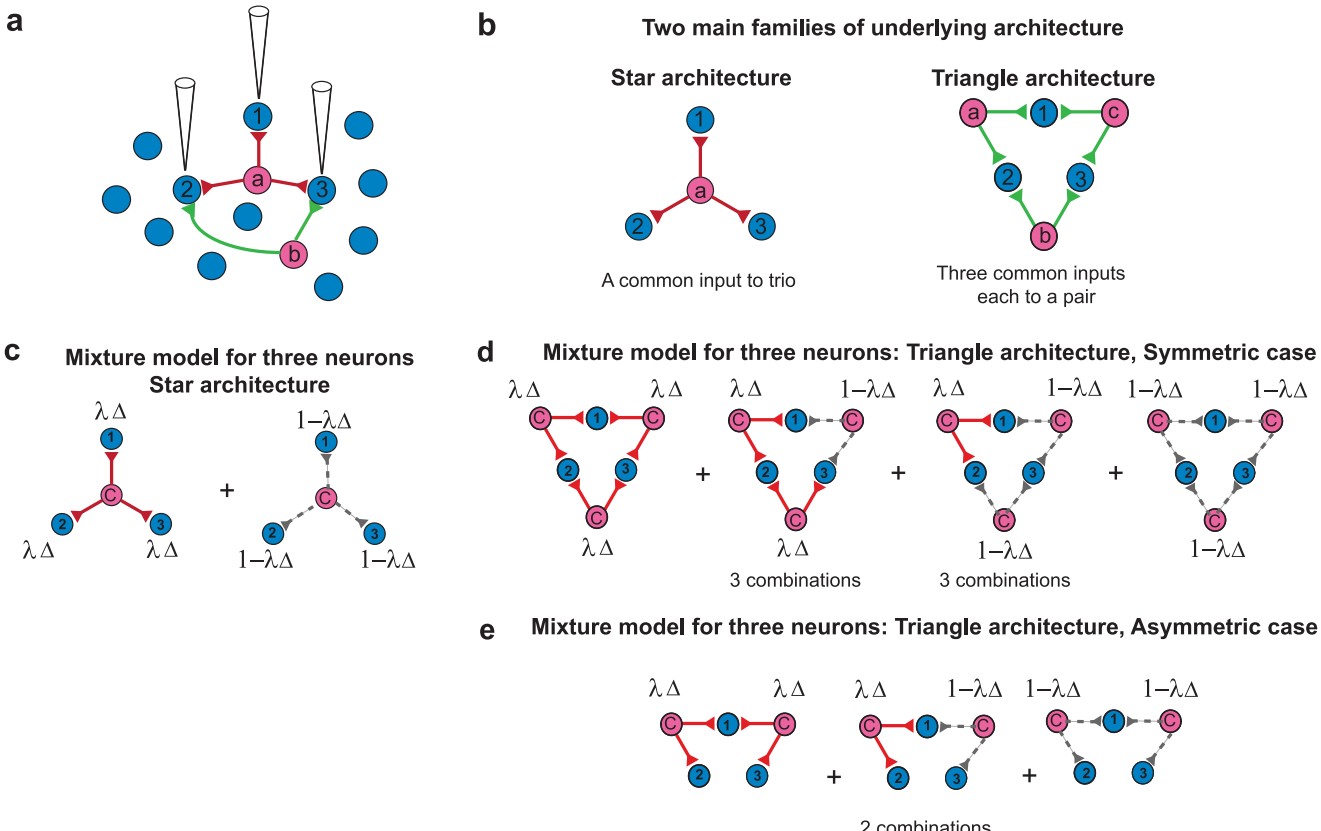

**Fig. 3 Schematic illustrations of the mixture models. a** Schematic diagram of simultaneous recording of three postsynaptic neurons. The neurons (blue circles) operate independently in the absence of any common input. The common input (pink circle) is to all three of them (red connections), or to two of them (green connections). **b** There are two main families of symmetric architectures: a common input to a trio (star architecture, left) and a common input to each pair out of three (triangle architecture, right). **c** Mixture model for three neurons in the star architecture where the neurons receive a common input (red lines) with probability $\lambda\Delta$ and do not receive it (gray dashed lines) with probability $1 - \lambda\Delta$. **d** Mixture model for three neurons in the symmetric triangle architecture with four distinct ways of combining the presence or absence of the common input (see the main text). **e** For asymmetric triangle architecture, the number of possible cases reduces to three.

The mixture models obtained by the occurrence probabilities are described in Methods (Methods: Mixture model of three neurons receiving common inputs to their pairs (triangle architecture)), including the asymmetric common input architecture in which there are only two common inputs out of three (asymmetric case) (Fig. 3e). We computed the triple-wise interaction of three neurons by using these mixture models.

For the two architectures above, we calculated the triple-wise interaction parameters from the simulated spike sequences of postsynaptic neurons and compared them with the theoretical predictions (Fig. 4a and b, left). The activities of neurons that receive simultaneous common excitatory input (star architecture) are characterized by positive triple-wise interactions (Fig. 4a, left). In contrast, the activities of neurons that receive independent common excitatory inputs to pairs (triangular architecture) are characterized by negative triple-wise interactions (Fig. 4b, left). Figure 4a, b (left) show that the triple-wise interaction decreases as the bin size increases for the same reason as in the pairwise interaction (Fig. 2b, right). The dependence of the triple-wise interaction on the common input rate is shown in the right panels of Fig. 4a, b.

Figure 4c shows the triple-wise interactions in the star (Top) and triangular (Bottom) architectures for excitatory (right) and inhibitory (left) common inputs as a function of the scaled diffusion coefficient (level of input noise) $D/(\tau_{\mathrm{m}} V_\theta^2)$ and scaled amplitude $A/(\tau_{\mathrm{m}} V_\theta)$ (see Supplementary Fig. 2b, c, e, f and Supplementary Note 1 for the simulation study). A single

common excitatory input in the star architecture (excitatory-to-trio) significantly increases the probability that all three neurons spike in the observation time window, $P(1, 1, 1)$, whereas a single common inhibitory input (inhibitory-to-trio) increases the probability of the reverse pattern, $P(0, 0, 0)$. The latter simply changes the sign of $\theta_{123}$ in Eq. (17) (Methods). In the triangular architecture with common excitatory input (excitatory-to-pairs), however, each common input causes postsynaptic spikes in two neurons but not in the other one. This primarily increases $P(1, 1, 0)$ (or a permutation of it) in the denominator of Eq. (17), resulting in negative $\theta_{123}$. For common inhibitory input in the triangular architecture (inhibitory-to-pairs), the probability of the reversed pattern, $P(0, 0, 1)$ (or a permutation of it) increases; this results in a larger numerator in Eq. (17) and positive triple-wise interaction. These results demonstrate that not only the type of common input (excitation or inhibition) but also the underlying architecture (star or triangular) determines the sign of the triple-wise interactions. We also observed that the magnitude of the negative triple-wise interactions induced by common inhibitory inputs is much weaker than those induced by common excitatory inputs. We expect this phenomenon will occur when the postsynaptic neuron exhibits a low spontaneous firing rate. We discuss why inhibitory inputs cannot generate strong interactions at low spontaneous firing rates in the section "Excitation versus inhibition: which one can produce stronger triple-wise interactions?" (see also Supplementary Fig. 3 and Supplementary Note 3).

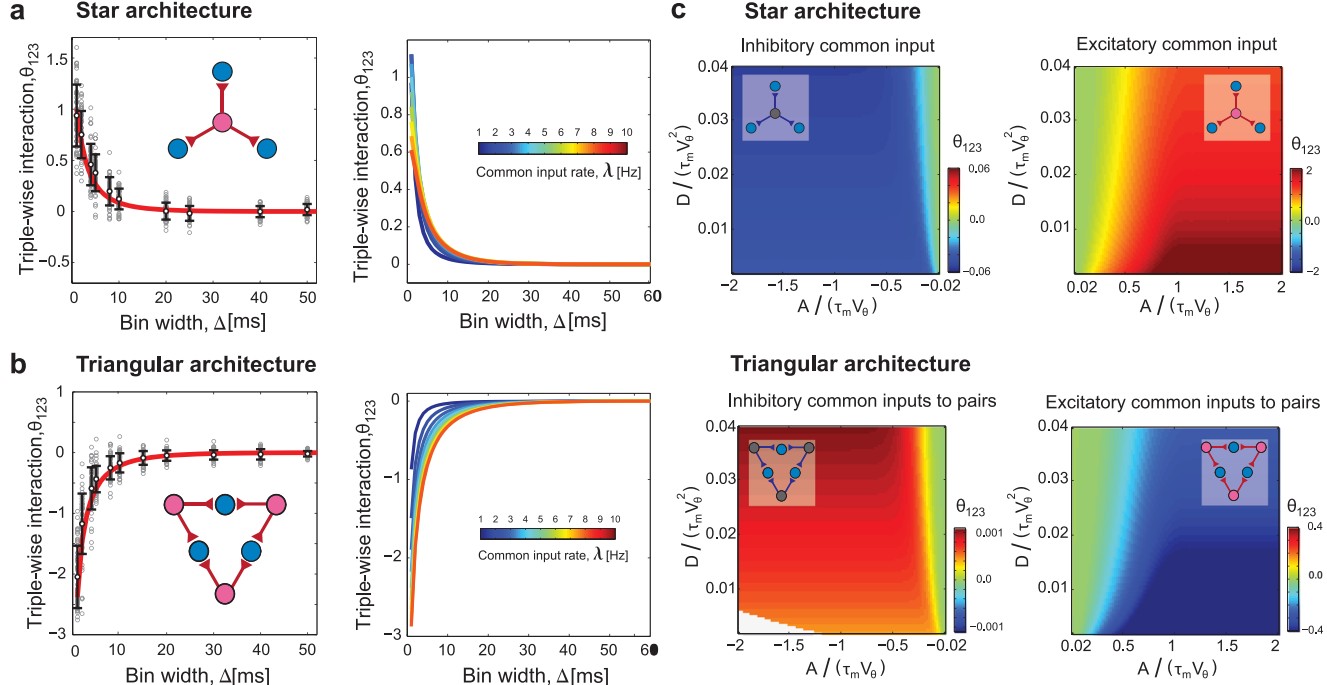

**Fig. 4 Triple-wise interactions among three neurons in the two leading architectures. a** The triple-wise interaction, $\theta_{123}$, in the star architecture: the three postsynaptic LIF neurons in blue simultaneously receive a common excitatory signal $A = 5$ mVms. The parameters are: $\tau = 20$ ms, $V_\theta = 20$ mV, $A = 5$ mVms, and $D = 0.74$ (mV)$^2$ms. Left: Triple-wise interaction computed from simulated spike sequences (gray lines: 50 individual trials each containing about 2500 spike occurrences from common inputs; dots and error bars: mean ± standard deviation) compared with the analytical result of the mixture model (red line, Eq. (17) in Methods) for $\lambda = 5$ Hz. Right: Analytical value of $\theta_{123}$ as a function of bin size, $\Delta$, for different common input rates, $\lambda$. **b** Triple-wise interactions in the triangular architecture: each pair of postsynaptic neurons in blue shares an independent common excitatory input. All parameters are as in **a**. **c** Triple-wise interactions among three neurons receiving common excitatory or inhibitory inputs in star (top panel) and triangular (bottom panel) architectures. In each row, the panel on the left is for common inhibitory and the one on the right is for common excitatory inputs. $\theta_{123}$ is a function of the scaled diffusion coefficient, $D/(\tau_m V_\theta^2)$, and scaled shared signal strength, $A/(\tau_m V_\theta)$; the other parameters are $\Delta = 5$ ms and $\lambda = 5$ Hz. The white region in the bottom left panel shows the numerically indeterminate region due to a very small diffusion coefficient (level of noise) and strong inhibition.

**Network structure and common input type can be determined from neuronal interactions.** The above observations raise a question: is it possible to determine the type of common input and the underlying architecture from the event activity of a neuronal population? Fig. 5a shows the first-order parameter, $\theta_1^t$, in a star or triangular architecture receiving either common excitatory or inhibitory inputs. Here, $\theta_1^t$ strongly depends on $D/(\tau_m V_\theta^2)$, which measures the level of the noise in background inputs, but only weakly depends on the signal's amplitude, $A/(\tau_m V_\theta)$. More importantly, it does not show any conclusive dependence, either on the choice of architecture or type of common input. Thus, it is impossible to identify the underlying architecture or the type of common input from the first-order parameters only.

However, the 2D plane of $\theta_{123}$ versus $\theta_{12}$ does differentiate motifs (Fig. 5b), as each motif occupies a distinct region. Thus, in principle, by investigating the interaction parameters, it is possible to identify the underlying architecture and type of common input (excitation or inhibition) to the three LIF neurons. However, within each motif (except for excitatory-to-trio), the parameters overlap, making it impossible to identify the underlying parameters such as the input's amplitude or diffusion coefficient from the interaction parameters. In addition, both the pairwise and triple-wise interactions are considerably weak when the neurons receive inhibitory inputs.

Each motif's boundaries in the $\theta_{123}$ versus $\theta_{12}$ plane are shown in Fig. 6a, right panel. The two inhibitory motifs occupying tiny areas are shown in the two panels on the left (top and bottom).

The three excitatory motifs cover much wider areas. Asymmetric excitatory-to-pairs motif is the other simple motif with shared excitatory inputs that can produce nonzero $\theta_{123}$. All five regions begin at the origin, $\theta_{12} = \theta_{123} = 0$ because both interactions vanish at zero signal amplitude.

We can explain the behavior of the neuronal interactions in Fig. 6a for the case of excitatory-to-trio motif as follows (see Supplementary Note 3 for the other architectures). Consider the dashed-dotted purple curve for postsynaptic neurons with a fixed spontaneous rate of $\mu = 1$ Hz; this curve shows how interactions change as one increases the shared signal's amplitude from zero to the highest conceivable value, i.e., $A/(\tau_m V_\theta) \gg 1$. The pairwise interaction monotonically increases with the signal strength and eventually saturates at its maximum value (the open black circle). The triple-wise interaction, however, shows non-linear behavior until it saturates. We can analytically show that, for any choice of the spontaneous firing rate $\mu$, we reach its corresponding saturation point at a sufficiently strong input amplitude (Supplementary Note 3). The saturation points (thick gray curve) are independent of the neuron model and the near-threshold assumption of the voltage, forming a universal upper boundary in the $\theta_{123}$ versus $\theta_{12}$ plane; the corresponding point for any of the excitatory-to-trio motifs is placed below it. To determine the lower boundary, we limited the spontaneous rates by $\mu \geq 1$ Hz. For any value higher than the background activity ($\mu > 1$ Hz), the corresponding curve appears above the curve for $\mu = 1$ Hz and below the universal upper boundary. Thus, the curve for $\mu = 1$ Hz acts as a practical lower boundary.

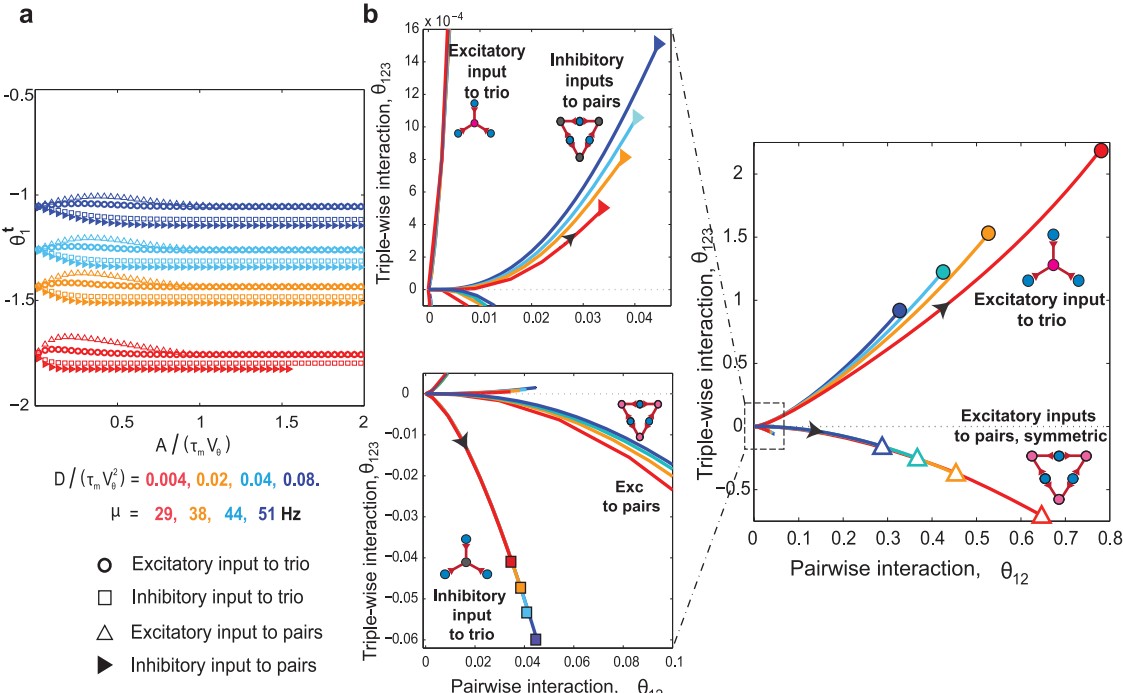

**Fig. 5 Natural parameters of population activity of three LIF neurons receiving common inputs in star and triangular architectures. a** The natural parameter for individual neurons, $\theta_1^t$, in star and triangular architectures with common excitatory or inhibitory inputs versus scaled shared signal strength $A/(\tau_m V_\theta)$. The symbols correspond to four motifs. The color codes are for four scaled diffusion coefficients. $\theta_1^t$ significantly varies with $D/(\tau_m V_\theta^2)$, which determines the postsynaptic spontaneous rate, $\mu$. However, $\theta_1^t$ does not show any significant dependence on the shared signal strength or any conclusive dependence on the type of architecture. **b** Triple-wise versus pairwise interactions for star and triangular architectures with common excitatory or inhibitory inputs as a function of scaled diffusion and scaled shared signal strength. The colors represent the levels of the scaled diffusion coefficient as in panel a. The black arrows indicate increasing directions of the scaled amplitude parameter (i.e., $A/(\tau_m V_\theta)$) from zero. The symbols at the end of the graphs show the saturation points of the interactions. The top and bottom panels on the left illustrate the interactions in the neighborhood of origin for negative and positive triple-wise interactions corresponding to inhibitory-to-trio and inhibitory-to-pairs motifs, respectively. The fixed parameters are the bin size, $\Delta = 5$ ms and presynaptic rate, $\lambda = 5$ Hz.

The stories for the other four motifs are similar to the one above. Each region contains numerous curves. To obtain each curve in a corresponding region, we set a certain postsynaptic spontaneous rate, $\mu$, and then vary the shared signal's amplitude from zero to a high value. This procedure yields a curve that begins at the origin and ends at its saturation point. Each region is the accumulation of these curves, and has two boundaries, one that is composed of all saturation points (thick gray boundary) and the other is the curve with the lowest firing rate $\mu = 1$ Hz (highest firing rate of $\mu = 100$ Hz), for motifs with excitatory (inhibitory) shared inputs (Fig. 6a and also Supplementary Note 4).

Experimentally verifying the analytically predicted regions (Fig. 6a) is challenging as it would require simultaneous recordings from the neurons and all their inputs. Instead, we used a multicompartmental neuron model in layer 5 of rat somatosensory cortex with a specific morphology from the blue brain project[3,57] to check the above theoretical predictions. We simulated each motif by adding shared inputs on top of other synaptic inputs received by three postsynaptic pyramidal neurons (NEURON simulator, Supplementary Note 5 and Supplementary Table 1). Figure 6b shows the resulting triple-wise versus pairwise interactions (mean ± 2 SD) of a simulation of the excitatory-to-trio (circles) and excitatory-to-pairs (squares) motifs while shared inputs have different amplitudes or efficacies (color code). The simulated results for each motif are placed within the predicted region. Interactions resulting from a shared input amplitude of 0.1 (dark blue circle and square) for both excitatory-to-trio and excitatory-to-pairs motifs cannot be distinguished from

interactions of the spontaneous activity (the black diamond at the origin). However, the excitatory-to-trio and excitatory-to-pairs motifs can be revealed by larger amplitudes of the shared input (from 0.2 to 4). As the amplitude of the shared input gets stronger, triple-wise and pairwise interactions in both motifs become stronger. In strong amplitudes of shared inputs, the data reaches the high amplitude line in excitatory-to-pairs motif (the red square at the bottom right). The simulation result of multicompartmental neuron for the inhibitory-to-trio and inhibitory-to-pairs motifs shows small and noisy pairwise and triple-wise interactions (red and blue, Supplementary Fig. 8) that could not be easily differentiated from spontaneous interactions (black circle, in Supplementary Fig. 8). These results confirm that our theoretical boundaries (regions) predict the architecture behind the activities of three multicompartmental neuron models.

Here we used the the log-linear model to trace the interactions and defined the boundaries of the motifs in the triple-wise versus pairwise interaction plane. To examine if it is a suitable measure, we investigated how well other measures of correlations, such as cross-correlation and covariance could distinguish the motifs (the definitions and calculations are in Supplementary Note 6). In particular, we found that the cross-correlation method (Supplementary Fig. 12b) produces boundaries for the excitatory inputs motifs that diverge for some range of spontaneous rate, and hence, it is not possible to identify motifs in that range (Supplementary Note 6 and Supplementary Fig. 10). On the other hand, the boundaries for the covariance measure are too narrow, and the correlations are too small (Supplementary Fig. 12c) to reliably distinguish among motifs. By comparison, the

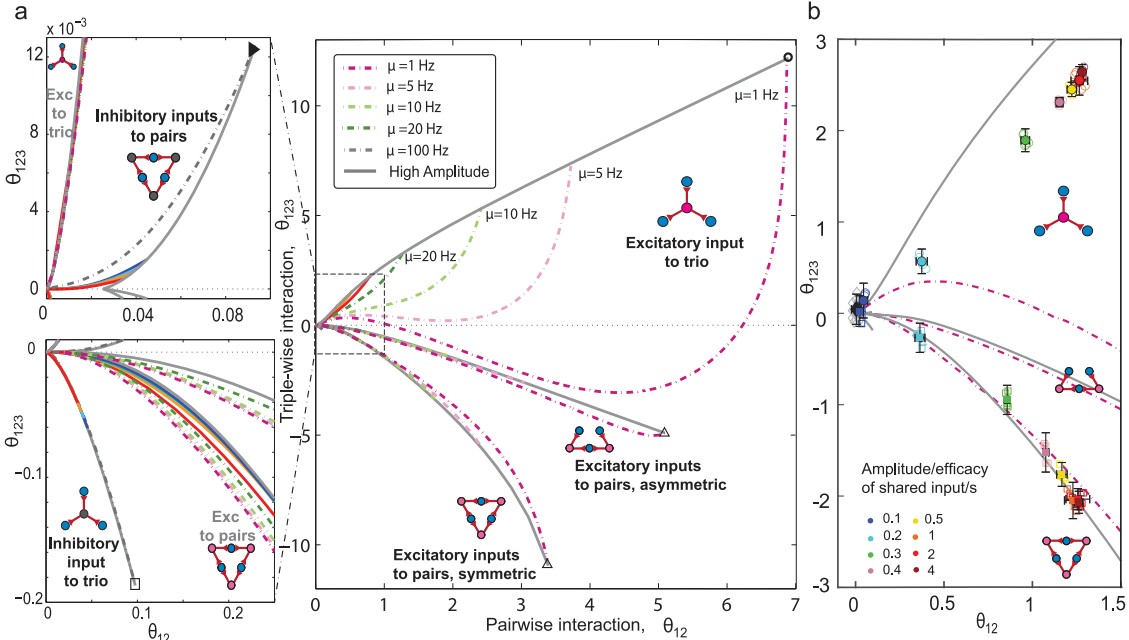

**Fig. 6 Distinct regions of interactions of three neurons' activity receiving common inputs of different architectures and synaptic types. a** Regions associated with, from top to bottom, excitatory-to-trio, asymmetric excitatory-to-pairs, and symmetric excitatory-to-pairs motifs. The regions associated with common inhibitory input are confined to an area near the origin (i.e., small $\theta_{123}$ and $\theta_{12}$), and are shown at higher resolution in the left panels. Each region is bounded by two analytical boundary lines: one at the high signal amplitude (solid gray lines); the other at the low or high spontaneous rate ($\mu = 1$ Hz or $\mu = 100$ Hz) for motifs with common excitatory or inhibitory input (the purple and gray dashed lines). The high amplitude limit is given by $|A|/(\tau_m V_\theta) \gg 1$; see Supplementary Note 3. The low (high) spontaneous rate limit corresponds to postsynaptic spontaneous rate of $\mu = 1$ Hz ($\mu = 100$ Hz); see Supplementary Note 4. We chose these limits to cover a wide range of postsynaptic spontaneous rates while maintaining the assumption of low activity rates: $\mu\Delta \simeq F_0(\Delta) \leq 0.5$; see Supplementary Note 3. For comparison, we added intermediate spontaneous rates: $\mu = 5, 10, 20$ (dashed lines). The colored solid lines (color code in Fig. 5) are numerical results of the LIF neuron model for different scaled diffusions and signal amplitudes (Fig. 5). The fixed parameters are the bin size $\Delta = 5$ ms, and the presynaptic rate, $\lambda = 5$ Hz. **b** In silico verification of theoretical regions using the multicompartmental neuron model of blue brain data[3,57]. The pairwise and triple-wise interactions of three neurons in excitatory-to-trio (circles) and excitatory-to-pairs (squares) motifs with different amplitudes of a 5 Hz Poissonian shared input (color code) are calculated from extensive simulation of specific pyramidal neurons in layer 5 of rat somatosensory cortex. Each plot (mean ± 2 SD) is the result of averaging over 5 runs of simulating 3 neurons for a 300s duration ($\Delta = 5$ ms; see Supplementary Note 5). The spontaneous interactions among three neurons without a shared signaling input is plotted as the black diamond near the origin.

interactions of the log-linear model (see Supplementary Note 6 and Supplementary Fig. 12a) do not have these difficulties, and the motifs can be distinguished. Therefore, the interaction parameters of the log-linear model constitute a better tool for identifying the hidden motifs behind correlations among neurons.

Finally, we clarify that which boundaries depend on the neuron model. We show that the high amplitude boundaries (thick gray curves) are independent of the neuron model, and the near-threshold assumption of the voltage (Supplementary Note 3). However, the other boundaries of the low (high) spontaneous rate for the excitatory (inhibitory) shared inputs show nontrivial behavior. For the star architectures, they remain independent of the neuron model, while for the triangle architecture, they depend on the choice of the neuron model and the near-threshold assumption (Supplementary Note 4). This dependence is an example of the nonlinearity of the input-output relation: it wouldn't exist if the probability of postsynaptic spike linearly increases with the strength of the presynaptic signal, i.e., an assumption for weak signals (technically, $F_A(\Delta) = F_0(\Delta) + const \times A$, Supplementary Note 4). In general, the probability of a postsynaptic spike varies non-linearly with the signal strength and saturates with strong signals; an accurate description of this dependence requires full knowledge of the neuronal model (Supplementary Note 4).

**How do more biological neuron models alter the model-dependent boundaries?** We analytically showed that the boundary curves of the triple-wise and pairwise interactions in Fig. 6a at high signal amplitude are independent of the neuronal model and the near threshold assumptions (see Supplementary Note 3). The universal boundaries also hold for curves of the excitatory and inhibitory-to-trio motifs. Therefore, a substantial portion, if not all, of the predictions would remain valid even if we change the neuron model. However, a more physiologically plausible model might modify the predicted interactions for excitatory-to-pairs motifs (i.e., low spontaneous rate boundaries). It is thus important to ask how much the obtained boundaries vary by changing the neuronal model or the near-threshold assumption of the voltage: can one region (area within two boundaries) entirely displace another, or even two distinct regions overlap?

First, we investigated how the model-dependent boundaries change by using more physiological neuron models other than the standard LIF neuron model. The LIF neuron model has certain limitations, e.g., in reproducing the variability of the inter-spike intervals observed in vivo[64–68]. To make it more biologically plausible, we added an adaptation term to the LIF neuron model[69–71] and simulated its effect on the pairwise and triple-wise interactions (see Supplementary Table 2 and Supplementary Note 7).

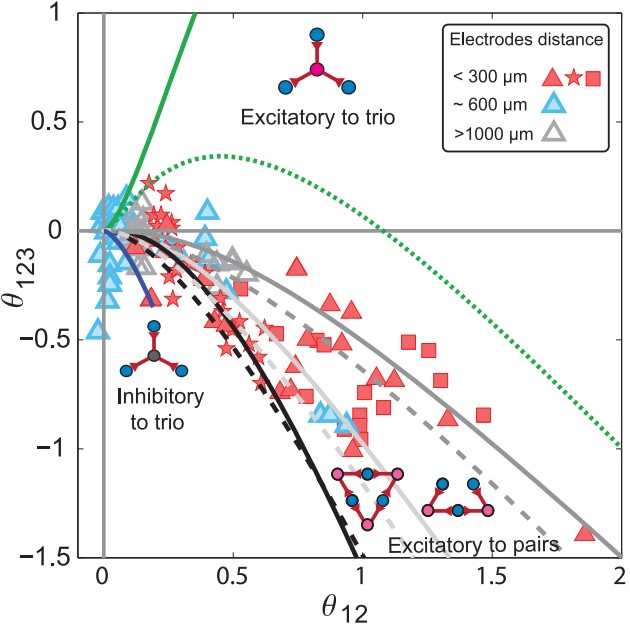

**Fig. 7 Comparison of theoretical predictions of neural interactions with V1 neurons of anesthetized monkeys.** Ohiorhenuan et al.[58] recorded spike data from V1 neurons of macaque monkeys by using tetrodes and analyzed the relation between triple-wise interactions (Eq. (17)) among three neurons (ordinate) and the average marginal pairwise interaction (Eq. (15)) of neuron pairs in the group (abscissa). The red and blue filled symbols represent interactions of neurons within 300 and 600 μm vicinity, respectively, while the unfilled gray symbol shows the interaction at >1000 μm distance (Replicated from Fig. 4 in Ohiorhenuan et al.[58]). The triangles are data from three neurons. The stars and squares are data from the four and five neurons. The black lines show the region of interactions for the triangular architecture with common excitatory inputs to each pair of neurons. The solid black line is the high amplitude boundary for interactions, while the dashed black line is the boundary of the low spontaneous rate ($\mu = 1$ Hz). The dark gray solid and dashed lines determine the boundaries for the asymmetric triangular architecture with common excitatory inputs given to two pairs among three neurons. The light gray lines are for the same asymmetric excitatory-to-pairs motif when the three pairwise interactions are averaged over, while the dark gray lines are the average of only nonzero pairwise interactions. The blue analytic lines represent a narrow region for the star architecture with common inhibitory input (high common input amplitude limit and high diffusion limit), whereas the green lines show the region for excitatory inputs given to three neurons (dashed line, low spontaneous rate boundary, $\mu = 1$ Hz for scaled diffusion limit of $D/(\tau_m V_\theta^2) = 2 \times 10^{-19}$ and solid line, high common input's amplitude limit). The time window is $\Delta = 10$ ms and common input rate is $\lambda = 5$ Hz.

The simulation results show that adaptation reduces the firing rate of the postsynaptic neurons (Supplementary Fig. 13 and Supplementary Fig. 14, Supplementary Note 7) in agreement with the experimental literature[72]. In the presence of adaptation, the common excitatory inputs generate even stronger pairwise and triple-wise interactions, while common inhibitory inputs induce weaker interactions (Supplementary Fig. 15).

The model-dependent low spontaneous rate boundaries for excitatory-to-pairs motif for the adaptive LIF (aLIF, Eq. S.47) and the adaptive exponential LIF (aEIF)[68,71] are shown in Supplementary Fig. 16; see also Supplementary Note 7. The high amplitude boundary remains the same because it is independent of neuron models. For the low spontaneous rate boundary in excitatory-to-pairs motifs, it can be seen that the more

generalized and biologically plausible aLIF and aEIF models preserve the region for each motif; the regions do not overlap.

Next, we examined whether the assumption of the near-threshold regime limited the validity of our results. We ran simulations of LIF neurons under subthreshold and suprathreshold regimes. The simulation results showed that common excitatory inputs produce stronger interactions in the subthreshold regime while common inhibitory inputs produce weaker ones compared with the threshold regime (Supplementary Note 8). The reverse happened in the suprathreshold regime (Supplementary Fig. 17). There is evidence that cortical neurons operate in the subthreshold (or near the threshold) regime[16] depending on the state of the animal or stimulus conditions[20], rather than in the suprathreshold regime that results in regular spiking. Consequently, when neurons are in the subthreshold regime, the inhibitory-to-trio region would become smaller and excitatory-to-pairs region would become larger, compared to Fig. 6a.

Section "Excitation versus inhibition: which one can produce stronger triple-wise interactions?" explains why the common excitatory inputs increase the higher-order interactions in the presence of adaptation or in the subthreshold regime and why these trends are reversed for common inhibitory inputs.

**Comparison of theoretical predictions with experimental data.** Figure 6 makes it possible to identify the underlying architecture and type of shared input (excitation or inhibition) for three homogeneous neurons by simply investigating their interaction parameters. As a practical example, we explored V1 neurons of anesthetized macaque monkeys studied in Ohiorhenuan et al.[43]. This study investigated the relationship between the triple-wise interaction (Eq. (17), Methods) of three neurons (ordinate) and the average marginal pairwise interactions (Eq. (15), Methods) of neuron pairs in the group (abscissa). The authors extracellularly recorded putative pyramidal neurons and found that many neurons, with mutual separations less than 300 μm, exhibited positive pairwise and strong negative triple-wise interactions (Fig. 7). The triple-wise interactions weakened when the electrode separation was increased beyond 600 μm. The authors speculated that the observed strong negative triple-wise interactions of the more nearby neurons are caused by the hidden activity of GABAergic inhibitory neurons, which presumably provide shared input to a large number of excitatory pyramidal cells[58]. The activities of V1 neurons were reported within 10–70 Hz[43,58,73], higher than the 1 Hz lower boundary, which we considered for the excitatory-to-trio motif. Thus, we can safely compare our theoretical predictions with the empirical observations. Figure 7 shows that the empirical data on most of the nearby neurons (filled red symbols) coincide with regions associated with the symmetric and asymmetric excitatory-to-pairs motifs. In contrast, neither the excitatory-to-trio nor any of the inhibitory motifs can explain most of the interactions of neurons within 300 μm. This clearly rules out the intuitive idea that shared inhibition could have induced the observed strong negative triple-wise interactions.

To see if the excitatory-to-pairs motif explains neuronal activity recorded from awake animals, we investigated neurons in the primary visual cortex of awake mice receiving drifting grating stimuli (Allen Brain Observatory—Neuropixels Visual Coding dataset)[59]. These neurons exhibit time-dependent firing rates (Fig. 8a). The distribution of firing rates of individual neurons are shown in Fig. 8b. To resolve the time-dependent firing rates in estimating their interactions, we fitted the state-space model of Eq. (16)[54,74], where we assumed the first-order parameters are time-dependent while the pairwise and

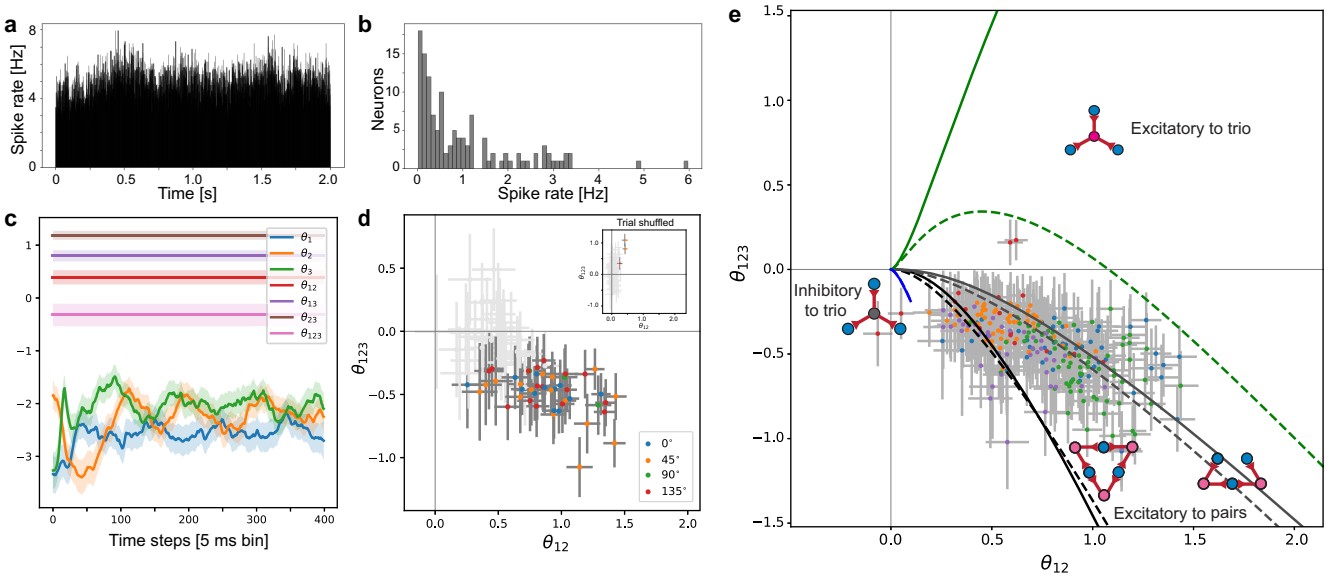

**Fig. 8 Comparison of theoretical predictions with V1 neurons of awake mice. a** Firing rate averaged over all trials and all V1 neurons recorded from an exemplary mouse during stimulus presentation. The stimulus was drifting gratings (orientation 0 degree, temporal frequency 2 Hz) with 75 repetitions in the 'functional connectivity' stimulus set of the Allen Brain Observatory—Neuropixels Visual Coding. The exemplary mouse had the largest recorded neurons in the dataset. **b** Distribution of firing rates of all recorded V1 neurons from the same mouse. **c** Exemplary traces of time-dependent and constant natural parameters estimated by applying the state-space method to recordings of activities of three neurons from the mouse' primary visual cortex (the solid lines are maximum a posteriori estimates; filled areas are 95% Bayesian credible intervals). The fitted model assumes that the natural parameters for individual neurons ($\theta_i$, $i = 1, 2, 3$) are time-dependent, while the pairwise and higher-order natural parameters are time-independent. The bin size used to create the binary data was 5 ms. The selected neurons had the top-three highest firing rates among the 126 recorded neurons of the mouse. **d** Triple-wise versus pairwise interactions. The pairwise and triple-wise interactions were estimated by independently applying the state-space analysis to all combinations of three neurons taken from the top-five highest firing rate neurons observed in the mouse with the largest number of recorded neurons. The stimuli are the drifting gratings with four different orientations. The colored dots are the MAP estimates of the interactions that significantly deviate from 0. Dots of same color are subsets of the data that were recorded under the same stimulus orientation. The error bars indicate 95% credible intervals. To draw the error bars for the pairwise interactions, we used the mean value of the estimated three pairwise interactions. We classified the subsets as significant if none of the 95% credible intervals of the pairwise and triple-wise interactions cover the value 0. **e** Comparison of interactions measured in five mice with theoretical boundaries for each architecture. We repeated the procedure to obtain panel **d** for the four other mice with the largest number of recorded neurons. The same color scheme of dots indicates the subsets of three neurons from the same mouse. The style and color of the boundaries follow the definitions used in Fig. 7. The gray lines are the asymmetric excitatory-to-pairs boundaries determined from the average of the nonzero pairwise interactions.

triple-wise interactions are time-independent (Methods: Sequential Bayesian estimation of the neuronal interactions). Figure 8c shows the estimated parameters for exemplary 3 neurons. The sequential Bayesian estimation method provides maximum a posteriori estimates (MAP) of each parameter with credible intervals. The model accounts for the modulation of firing rates with the time-dependent first-order parameters.

Next, we fitted this model to all the combinations of five neurons exhibiting the highest firing rates. Figure 8d shows MAP estimates of the pairwise and triple-wise interactions of these neurons. The error bars are the 95% credible intervals. The colored dots and darker error bars indicate that the interactions are significantly away from 0; namely, if the minimum values of the 95% credible intervals of all pairwise and triple-wise interactions are larger than 0 or if the maximum values of the 95% credible intervals are all smaller than 0. The plots in light gray are non-significant groups of neurons. The directions of the grating stimulus are indicated by different colors of the dots. Note that the significant positive pairwise and negative triple-wise interactions are not the artifacts of the time-dependent modulations (see the inset for the trial-shuffled result).

Finally, we repeated the same analysis on the five mice with the largest number of recorded neurons. Figure 8e is a summary plot with the theoretical boundaries for the five mice (only the groups showing significant interactions are shown). The results of the time-dependent analysis with credible intervals are consistent

with the anesthetized monkey data: the excitatory-to-pairs motif is the most plausible for explaining the observed neural interactions.

The above findings trigger the following question: why should the observed strong negative triple-wise interaction be associated with common excitatory inputs, and the inhibitory shared inputs fail to produce any strong negative interactions? We will answer this question at the end of the Results section. But before that, we verify the robustness of our excitatory-to-pairs scenario to a possible complication of the motifs by recurrent interconnections.

**Excitatory directional/recurrent connections among three neurons can explain the observed negative triple-wise interaction.** Although the previous section's analysis seems valid for common input architectures among three postsynaptic neurons, the question remains as to whether considering interconnections among the three neurons might affect our conclusions from Figs. 7 and 8. Therefore, we simulated all possible motifs of directional or reciprocal connections among the three neurons. The number of motifs for three neurons is $2^6 = 64$ (each directed connection can be present or absent; $2^6$ for the 6 possible interconnections). However, some of these motifs are structurally the same: they turn into each other simply by permuting the labels of the three postsynaptic neurons. This means that the 64 possible motifs reduce to 16 main structures (Fig. 9a). Here, we asked

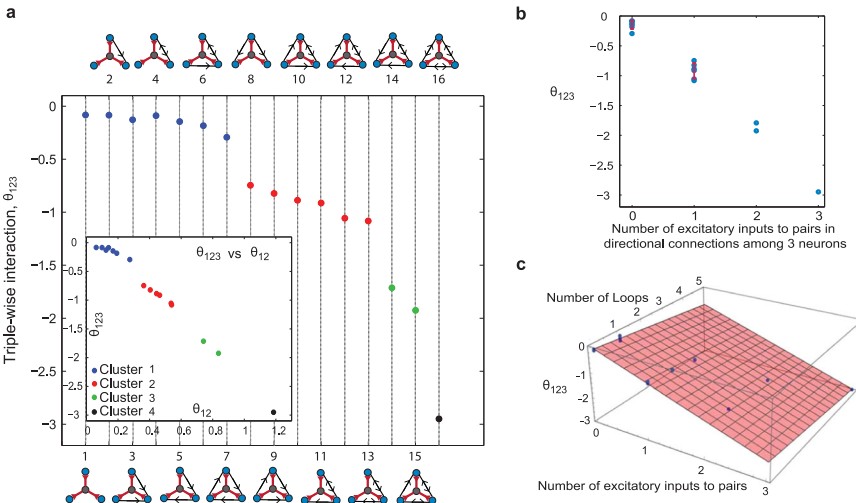

**Fig. 9 Triple-wise interactions for 16 motifs of directional and/or reciprocal connections among three neurons when they receive independent noise and common inhibitory input to trio. a** The architectures are divided into four clusters based on the number of excitatory inputs to pairs in each motif. The first cluster (blue) contains directional connections with no excitatory inputs to pairs in the directional connections among three neurons (blue circles). The second cluster (red) has one excitatory input to a pair, and the third and fourth (green and black) are related to two and three excitatory inputs to pairs in the directional connections among three neurons (blue circles). The inset shows the triple-wise interaction versus pairwise interaction for all 16 motifs. **b** The mean and error bar (standard deviation) of triple-wise interaction for each cluster as a function of the number of excitatory inputs to pairs. **c** Triple-wise interaction for the 16 motifs is a linear function of the number of excitatory-to-pairs motifs in the directional connections and number of loops. Each motif was simulated 50,000 times, and each trial contained 500 seconds of spike trains with a time resolution of 0.05 ms. The time window used to calculate the triple-wise and pairwise interactions was $\Delta = 5$ ms, and the shape of presynaptic input was a square function for both the common input and directional connection's input, the same as in the analytic calculation. Scaled diffusion $D/(\tau_m V_\theta^2) = 9 \times 10^{-5}$; scaled amplitude $A/(\tau_m V_\theta) = 0.0125$; common presynaptic input and input rate of directional connectivity $\lambda = 5$ Hz; the time delay of directional interconnection $t' = 6$ ms $> \Delta$; $A'/A = 1$, where $A'$ is the amplitude of the directional connections.

whether adding directional or reciprocal connections between neurons in the inhibitory-to-trio motif would shift the strength of triple-wise interactions arising from inhibitory-to-trio toward the strong interactions found in the experimental data (for example, the red symbols in Fig. 7). Figure 9a shows the results of triple-wise interaction for each motif averaged over 50,000 runs. We found four clusters of motifs, separated from each other, ordered by the number of inputs to pairs. The first cluster (blue, motifs 1 to 7) contains motifs with no simultaneous input from one excitatory neuron to two others (to pairs), despite that there can be recurrent connections. The average triple-wise interaction induced by this cluster is small. The second cluster (red, motifs 8 to 13) has motifs with one excitatory neuron as the input with directional connections to pairs of neurons. The third and fourth clusters (green, motifs 14 and 15; and black, motif 16) contain motifs with two and three excitatory-to-pairs of neurons in the directional connections between neurons. Clearly, as the number of excitatory inputs to pairs increases, the triple-wise interaction becomes more negative and stronger. The inset shows the triple-wise versus pairwise interactions for these four clusters. While motifs 2-7 in the first cluster (blue) can not be distinguished from the inhibitory-to-trio (motif 1), the strong negative triple-wise interactions observed in the second, third, and fourth clusters (motifs 8–16) cannot be explained by the inhibitory motif (motif 1) even if the amplitude of the inhibition is increased (Fig. 4c). This picture is consistent with the empirical data (Figs. 7 and 8), where there were large negative triple-wise and positive pairwise interactions. The excitatory-to-pairs motif, either as common input or as a directional connectivity, can generate strong interactions and thus is supported as the basic motif behind the data presented in Figs. 7 and 8.

Another question remains about the simultaneous existence of common excitatory and inhibitory inputs in both the triangle and star architectures; in this case, we analyzed a model in which these architectures were mixed (Supplementary Note 9). The

results show that mixing of other motifs excluding the excitatory-to-pairs motif cannot induce strong negative triple-wise and positive pairwise interactions (Supplementary Fig. 18).

**Excitation versus inhibition: which one can produce stronger triple-wise interactions?** So far, we found that the strong negative triple-wise combined with positive pairwise interactions observed for V1 neurons are a signature of microcircuits with common excitatory inputs (Fig. 7). One remaining question is why other microcircuits with common inhibitory inputs failed to produce strong negative triple-wise interactions. Another is can we always attribute strong higher-order interactions to common excitatory inputs, or does it depend on certain features which vary from experiment to experiment?

The measured pairwise and triple-wise interactions depend on various features of the postsynaptic neurons, as well as their possible shared inputs. For the analytically tractable regime of strong signaling inputs, however, we can reduce many factors to a few decisive ones. Here, analytical calculations show that, when the spontaneous rate of postsynaptic neurons in the time window, $\Delta$, is low, i.e., $F_0(\Delta) \ll 1$, common excitatory inputs produce large pairwise and triple-wise interactions, while common inhibitory inputs do not (Supplementary Fig. 3 and Supplementary Note 3). This picture is reversed if the spontaneous firing rate of the postsynaptic neurons is high, i.e., $F_0(\Delta) \lesssim 1$. There is, of course, an intermediate regime, $F_0(\Delta) \simeq 0.5$, where the strength of the interaction induced by inhibitory inputs to trio and excitatory inputs to pairs are nearly the same (Supplementary Fig. 3).

Figure 10 illustrates how the postsynaptic neurons' spontaneous rate, i.e., $\mu$ within the time bin, $\Delta$, plays an essential role in relating the hidden underlying architecture with the observed interactions. If the regime of spontaneous rate is known, based on the statistics of neural data (pairwise and triple-wise interactions), one can predict the predominant architecture that induces the observed interactions. In the low spontaneous rate regime,

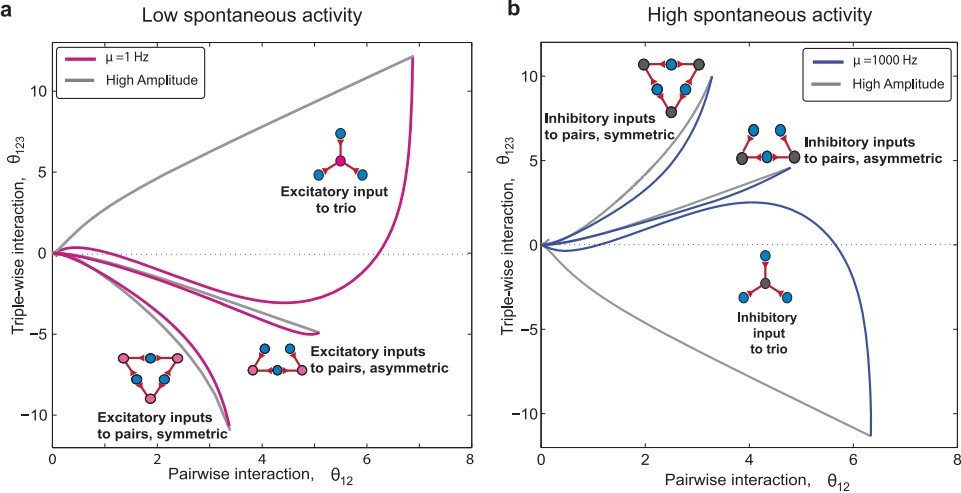

**Fig. 10 Guide map for uncovering the underlying architecture from observed higher-order interactions. a** For a spontaneous rate within the range $\mu = 1 - 100$ Hz, three regions for excitatory inputs' motifs are shown. The strong higher-order interactions, regardless of their sign, are the sole signature of common excitatory inputs when the spontaneous rate is low. **b** The regions for spontaneous rate within the range of $\mu = 100 - 1000$ Hz show that inhibitory inputs can induce strong interactions in a high spontaneous rate regime. The excitatory input regions for high spontaneous rates (Supplementary Fig. 4) shrink to a small size compared with the low spontaneous rate regime. The method of drawing the boundaries follows Fig. 6a and Supplementary Note 3 and 4. The fixed parameters are the bin size, $\Delta = 5$ ms, and input rate, $\lambda = 5$ Hz.

motifs of excitatory inputs can induce strong triple-wise and pairwise interactions (regions in Fig. 10a); whereas, in the high spontaneous rate regime (for example, in the olfactory bulb[75]), motifs with inhibitory inputs can generate strong interactions (Fig. 10b). When judging the architecture, it is recommended to consider uncertainty in estimating the empirical interactions to avoid erroneous detection of the architecture.

In the experiment conducted by Ohiorhenuan et al.[43,58], the neuronal firing rates ranged within $10\,\text{Hz} \le \mu \le 70\,\text{Hz}$ (Fig. 4 in ref. [58]), while the time bin was $\Delta = 10$ ms. The exact spontaneous spiking probability of postsynaptic neurons is $F_0 = 1 - \exp(-\mu \times \Delta)$; this yields $0.1 \le F_0 \le 0.5$. For such values of $F_0$, any observation of strong triple-wise interactions is an indication of common excitatory inputs as opposed to inhibitory ones. For the V1 neurons of mice, we chose neurons that had firing rates in $3\,\text{Hz} \le \mu \le 6\,\text{Hz}$, a time bin of $\Delta = 5\text{ms}$, and spontaneous spiking probability ranges within $0.015 \le F_0 \le 0.03$. As such, the observed strong negative triple-wise interactions are a signature of excitatory-to-pairs motifs. Furthermore, the data from Ohiorhenuan and Victor[58] indicates that neurons exhibiting lower firing rates generate stronger positive pairwise and negative triple-wise interactions (see Fig. 4c, a in Ohiorhenuan and Victor[58]). This observed increase in the pairwise interactions and negative triple-wise interactions with decreasing spontaneous rate (firing rate) appears in excitatory-to-pairs motif in Supplementary Fig. 3a, b (right), but not in inhibitory-to-trio motif (Supplementary Fig. 3a, b, left). This fact reaffirms our claim that the excitatory-to-pairs motif underlies the V1 microcircuits in these datasets.

## Discussion
Our results point to the possibility of revealing the underlying neuronal architecture and type of common input by using pairwise and triple-wise neural interactions (Fig. 10). Furthermore, for the specific set of monkey and mouse data, we conclude that, rather than the inhibitory-to-trio motif, the excitatory-to-pairs motif, either as hidden common inputs or directional connection, is a necessary and sufficient explanation of the observed strong negative triple-wise and positive pairwise interactions. For revealing the motif underlying the data, we presented an analytic

guide map: showing the distinct regions of each basic motif in the triple-wise versus pairwise plane (Fig. 6a). Each region is defined by two boundaries, the high amplitude regime boundary for all motifs and the low (high) spontaneous rate boundary for the excitatory (inhibitory) inputs motifs. For the high amplitude boundary, we analytically calculated how extremely strong common inputs affect the pairwise and triple-wise interactions (Supplementary Note 3). This analysis was independent of the neuron models and the conditions of the equilibrium potential: it reveals that whenever the spontaneous firing rate is low, motifs that have excitatory inputs can induce strong triple-wise interactions (Supplementary Fig. 3), whereas when the spontaneous firing rate is high, motifs with common inhibitory input can produce strong interactions. A neuron in this situation may resemble one talkative person (excitatory input) who is clearly noticed among many others who are silent (low spontaneous rate). On the other hand, if the majority are talkative (high spontaneous rate), one silent person (inhibitory input) would be conspicuous.

We showed that low (high) spontaneous rate boundary for the excitatory-to-trio (inhibitory-to-trio) motif as well as high amplitude boundaries for all motifs, are independent of the neuron model and the near-threshold assumption (Fig. 6a). The low (high) spontaneous rate boundary of the common excitatory (inhibitory)-to-pairs, however, depend on the neuron model and the near-threshold assumption (Supplementary Note 4). We carried out numerical simulations and other verifications to make sure (i) the observed strong negative triple-wise interactions and positive pairwise interactions in particular data on macaque and mouse V1 are signatures of the excitatory-to-pairs motif and (ii) the guide map remains reliable, in more general situations like including adaptation in the neuron model (Supplementary Note 7) and when the neuron's voltage is slightly away from the threshold (Supplementary Note 8). However, the guide map and regions corresponding to motifs changed according to the spontaneous rate of neurons and bin size. Here, we have provided the results for infrequent and frequent spontaneous rate of postsynaptic neurons (Fig. 10) under the assumption of a low input rate and small bin size. The dependence of guide map on bin size is shown in Supplementary Fig. 5. As can be seen, the bin

size cannot be too large, as it would diminish the effect of the shared input (Fig. 4a, b) and hence degrade the overall reliability of our formalism.

The classical approach to infer synaptic connectivity from extracellular spiking activity is to construct cross-correlograms of simultaneous spike trains from pairs of neurons[10]. However, this approach aims at discovering connections among recorded neurons. In fact, researchers have made efforts to eliminate the effect of common drives from unobserved inputs on this measure to avoid erroneously reporting pseudo-connections[11,76]. Another approach is the model-based method that uses a stochastic model of neurons. Among them, the point process-generalized linear model (GLM) is a standard tool for analyzing the statistical connectivity of the observed neurons[12–14]. However, these models determine current neuronal activity from their own past activities and/or known covariate signals such as stimulus and local field potential signals. Since the recorded neurons are embedded in larger networks, we need to consider the effect of inputs from unobserved neurons to describe the population activity accurately. Although there have been attempts to incorporate common inputs from unobserved neurons into the GLM framework by treating them as hidden variables[77,78], these statistical models are not directly constrained by physiological limits such as Dales' law, physiological membrane dynamics, or spiking thresholds that the LIF neuron model has[79]. In contrast, the physiological LIF models that we used are based on knowledge about the balanced network: we included hidden inputs as background noise and consider various architectures of hidden common inputs as shared signals with arbitrary strengths. Moreover, we generalized the analysis and presented a guide map to infer motifs with most of its boundaries independent of the neuron model.

Another approach to model the input-output relation of a neural population under in vivo conditions is to use the dichotomized Gaussian (DG) model[47,50,80] and its extensions[81–84]. The DG model contains threshold functions that receive inputs sampled from a correlated multivariate Gaussian distribution to model shared synaptic inputs. It can capture well the population spike-count distributions of exponential integrate-and-fire neurons receiving shared Gaussian inputs[85]. Previous studies have shown that this simple model exhibits positive pairwise and negative triple-wise interactions and can account for the observed sparse population activity[41,42]. However, our approach to model input-output relation has the following advantages over the DG model. First, we describe the population activity based on the analytical input-output relation of the LIF model, as opposed to the DG model that lacks membrane dynamics. The main merit of the DG model is that, due to the simplified construction without dynamics, it offers an analytical expression of the population activity given the statistics of the inputs. However, we recently obtained the analytic input-output relation for the near-threshold LIF neuron that addresses the dynamics of the in vivo membrane potential[36], which allows us to describe the input-output relation with greater temporal accuracy. Second, the LIF neuron models allow us to construct various architectures with different common input types, enabling us to build a flexible framework to infer the network structure from data. In contrast, the DG model is limited in its structure of the shared inputs; thus, one cannot test alternative hypotheses, e.g., whether common inhibitory inputs can also generate the same neural interactions[41,58].

Our quantitative model is based on two distinct network architectures (triangle and star) with either excitatory or inhibitory shared inputs. It is crucial to determine how the directional connections among postsynaptic neurons affect the prediction. Ample experimental evidence has established that pyramidal neurons in the visual cortex of mature mammals are sparsely connected[22,86–89]. However, a combination of directional connections with shared inputs has been observed. For example, excitatory inputs from layer 4 are shared with layer 2/3 connected pairs of excitatory pyramidal neurons[90]. Here, we ran simulations on directional connections among three neurons in addition to inhibitory input to trio: the results affirm that the presence of excitatory-to-pairs motif, rather than inhibitory-to-trio, either in directional or hidden shared input's connections induces strong negative triple-wise and positive pairwise interactions in low spontaneous rate regimes (see Fig. 9). In some experimental results, of course, a divergent common inhibition might be mixed with local common excitatory inputs. To find evidence of such a mixed inhibitory-to-trio motif in the data, one should carefully examine deviations from the observed interactions that are solely due to the excitatory-to-pairs motif. However, we expect such deviations would be small, as long as the spontaneous activities of neurons are low.

One of the assumptions in our analytical framework is that the firing rate of the signaling input is low[91] in comparison with that of the postsynaptic neurons; Hence, there is at most one signal arriving between two successive spikes of the postsynaptic neuron. It is possible to consider cases with higher firing rates of the signaling input (Appendix III in Shomali et al.[36]). However, as we have considered a small time window, $\Delta = 5 - 10$ ms, the assumption of having not more than one signal during such a short time window is reasonable (Supplementary Fig. 2 in Supplementary Note 1). Another assumption is that the synaptic inputs set the voltage of the neuron near the threshold regime, which is reported to be the case when stimulus is presented[20]. However, a verifying analysis by NEURON simulator using the blue brain multicompartmental neuron model[3,57], supports the idea that the guide map remains intact even in subthreshold situations.

Our finding shows that the strong negative triple-wise interactions (sparse population activity) observed in the data[43] can be induced by a simple motif, the excitatory-to-pairs, with the realistic spiking nonlinearity. Does this microcircuit have any specific computational advantage, or did the specific experimental settings result in this observation? Independent empirical evidence shows that the excitatory-to-pairs motif is overexpressed compared with a random network in rat visual[23] and somatosensory[92] cortex. Our findings of excitatory-to-pairs motifs in monkey and mouse V1 neurons may imply that such ubiquitous motifs coupled with neurons' nonlinearity are sufficient for sparse coding in the early sensory cortices[93–95]. A theoretical study by Zylberberg and Shea-Brown[53] supports this view: they showed that, when recurrently connected neurons optimally encode natural images, they sparsified the population responses by integrating inputs sublinearly (i.e., exhibited negative triple-wise interactions). On the other hand, the common inhibitory input traditionally plays a role in the sparse coding in the form of a well-known winner-take-all network[96]. Indeed, there is also experimental evidence that a common inhibitory input innervates multiple postsynaptic pyramidal neurons if they are closer than $100\,\mu m$[97] to each other; at greater distances, the probability of common inhibitory inputs to two or more neurons decreases (Fig. 6b in Packer and Yuste[97]). This phenomenon is attributed to the limited length of the inhibitory neurons' axons and it means that, if the electrodes' separation is greater than $100\,\mu m$, the chance of capturing a common inhibitory input (to a pair, or to a trio) is diminished. In the experiment of Ohiorhenuan et al., the closest possible separation of the recorded neurons was less than $300\,\mu m$; i.e., recorded neurons are expected to be in a circle of radius $r \sim 150\,\mu m$[58]. In that case, the probability of having two neurons closer than $100\,\mu m$ is 32%, while that of having three neurons closer than $100\,\mu m$ to each other would be much lower, less than 7% (Supplementary Note 10). Thus, the

probability of finding an inhibitory presynaptic neuron, that innervates synapses to three recorded postsynaptic neurons is less than 7%. Consequently, the observations of Ohiorhenuan et al. do not rule out the presence of an inhibitory-to-trio motif in local microcircuitry spanning length of less than 100 μm, and it cannot be used as empirical proof that the excitatory-to-pairs is an exclusive computational motif of the V1 microcircuits[58].

More precise experiments controlling the separation of electrode tips are required to complete the picture of cortical microarchitecture. Plotting how the observed triple-wise interaction varies with neuronal separation might give a clearer picture. Such a dependence of pairwise interactions varied with neurons' distance has been observed in retina ganglion cells[98]. If the chance of a strong negative triple-wise interaction for neurons closer than 100 μm decreases, it would indicate the absence of the excitatory-to-pairs motif in a local network with neuronal separation less than 100 μm; if so, Ohiorhenuan et al.'s observation would be a specific result of the experimental setting[58]. However, if the strong negative triple-wise interactions persist even for neurons closer than 100 μm, it means that the excitatory-to-pairs motif prevails in the microcircuit ($\leq$ 300 μm) and would constitute further evidence supporting the computational advantage of excitatory-to-pairs microcircuitry. By contrast, it would be difficult to find evidence that the inhibitory-to-trio motif exists or coexists with the excitatory-to-pairs motif as computational units in the local microcircuits from the activities of the three neurons as long as the postsynaptic firing rate is low, because of the small negative triple-wise interactions induced by common inhibitory inputs (Supplementary Note 3).

Finally, we hope that the experimental evidence of the structured interactions in mouse V1 (Fig. 8) and theoretical predictions of the underlying motifs for both mouse and monkey V1 data presented here, will motivate neurophysiologists to perform experiments that can directly identify input types and the network's structure in living animals. More specifically, although it is quite challenging, an experiment that simultaneously performs in vivo patch-clamp of postsynaptic neurons and common inputs (and specifies the types of neurons by using, e.g., genetic methods) could provide the ground-truth data about the architecture and improve the prediction of the proposed method.

In summary, we provided a theoretical tool and verified it by NEURON simulator, to predict network architecture and types of hidden inputs (excitatory/inhibitory) from the activity of neurons recorded in vivo. We defined analytic regions for each motif, with boundaries mostly independent of a neuron model, to show that the basic motifs can be distinguished using spiking statistics. Our guide map helps to uncover hidden network motifs from neural interactions observed in a variety of in vivo data.

## Methods

### Effect of presynaptic spike-timing on leaky integrate-and-fire neuron receiving noisy inputs balanced near threshold.
Here, we describe the statistical properties of our cortical neuron model operating under in vivo-like conditions[36]. We evaluated the probability of spiking within a given time window; it is the building block with which we construct the population activity of such neurons. To this end, we use a leaky integrate and fire (LIF) postsynaptic neuron with a membrane time constant of $\tau_m$ and a resting potential of $V_r$:

$$\tau_m \frac{dV(t)}{dt} = -(V(t) - V_r) + I(t).$$ (5)

The neuron spikes when its membrane potential, $V(t)$, hits the spiking threshold, $V_\theta$; $V(t)$ then resets to $V_r$. The input current $I(t)$ consists of two parts: (a) a transient signaling input which represents the input from the influential synapses of arbitrary strength, $\Delta I(t, A, \tau_b)$, and (b) the effect of all other independent presynaptic inputs accumulated as a fluctuating background input, $I_0(t)$:

$$I(t) = I_0(t) + \Delta I(t, A, \tau_b).$$ (6)

We modeled the fluctuating background input as Gaussian white noise, so that it would replicate synaptic inputs to V1 neurons when a visual stimulus is

presented[20]. $I_0(t)$ has a mean drive of $\bar{I}$ and variance of $2D/\tau_m$; here, the diffusion coefficient, $D$, measures $I_0(t)$'s level of noise. The signaling input, $\Delta I(t, A, \tau_b)$, is characterized by its amplitude (or efficacy), $A$, and its arrival time, $\tau_b$.

The fluctuating $I_0(t)$ is a source of variability; its stochastic nature, however, makes it impossible to solve Eq.(5) and find the exact spike-time deterministically. Thus, researchers have tried to address the probability of spiking[36,68,99]. Their essential mathematical tool is the Fokker-Planck (or diffusion) equations[100], which give the probability density that a postsynaptic neuron spikes at time $t$, given that the membrane potential at the initial time $t_0$ is known. However, the corresponding Fokker-Planck equation has yet to be solved even in the absence of any signaling input. An analytical solution exists for a very specific case $\bar{I} = V_\theta$[101,102], which is known as the threshold regime representing a physiologically plausible situation for in vivo neurons. Recently, Shomali et al. were able to extend that analytic solution of the spike density to a case in which signaling inputs arrive on top of background noise[36]. They considered a near-threshold neuron, $\bar{I} \simeq V_\theta$, that receives a transient signaling input (i.e., the synaptic time constant of $\tau_s$ is sufficiently smaller than the membrane time constant, $\tau_m$). They solved the Fokker-Planck equation and analytically found the probability density of spiking (also called the inter-spike interval distribution, or ISI) for an arbitrary strength and shape of signaling input (Methods: Spike density of a leaky integrate-and-fire neuron receiving a signaling input at the threshold regime).

### Spiking density of leaky integrate-and-fire neuron receiving signaling input in the threshold regime.
According to Shomali et al.[36], the first-passage time density (inter-spike interval density) for the LIF neuron (Eq. (5)) receiving signaling input at time $\tau_b$ on top of noisy background input can be expressed as

$$J_A(t) = \frac{\sqrt{\kappa\omega}}{\pi\tau_m}\left\{\exp\left(-\frac{\varphi_+^2}{2}\right)\left[1 + \sqrt{\frac{\pi}{2\kappa}}\varphi_+ \exp\left(\frac{\varphi_+^2}{2\kappa}\right) \times \left(1 + \mathrm{Erf}\left(\frac{\varphi_+}{\sqrt{2\kappa}}\right)\right)\right] \right.$$
$$\left. - \exp\left(-\frac{\varphi_-^2}{2}\right)\left[1 + \sqrt{\frac{\pi}{2\kappa}}\varphi_- \exp\left(\frac{\varphi_-^2}{2\kappa}\right) \times \left(1 + \mathrm{Erf}\left(\frac{\varphi_-}{\sqrt{2\kappa}}\right)\right)\right]\right\},$$ (7)

where $\mathrm{Erf}(x) = (2/\sqrt{\pi})\int_0^x \exp(-t^2)dt$ and $\kappa(t, \tau_b)$, $\omega(t, \tau_b)$ and $\varphi_\pm(t, \tau_b)$ are

$$\kappa(t, \tau_b) = (1 - r(t)^2)/(1 - r(t - \tau_b)^2),$$

$$\omega(t, \tau_b) = \frac{r(t - \tau_b)^2(1 - r(\tau_b)^2)}{(1 - r(t)^2)^3},$$

$$\varphi_\pm(t, \tau_b) = \sqrt{\frac{\tau_m V_\theta^2}{D(1 - r(\tau_b)^2)}}\left\{\pm r(\tau_b) - \int_{\tau_b}^t \frac{ds}{\tau_m}\frac{\Delta I(s, A, \tau_b)}{V_\theta}\right\},$$ (8)

using $r(t) = \exp(-t/\tau_m)$. The first-passage time density in the period before the signal arrival, i.e., $t < \tau_b$, reduces to the known formula[102,103]:

$$J_0(t) = \frac{1}{\tau_m}\sqrt{\frac{2}{\pi}\frac{\tau_m V_\theta^2}{D}\frac{r^2(t)}{(1 - r^2(t))^3}}\exp\left[-\frac{\tau_m V_\theta^2}{2D}\frac{r^2(t)}{1 - r^2(t)}\right].$$ (9)

In this study, we used Eq. (7) with a square-shaped signaling input given by:

$$\Delta I(t, A, \tau_b) = A \times \begin{cases} 0 & t < \tau_b, \\ 1/\Delta t & \tau_b \leq t \leq \tau_b + \Delta t, \\ 0 & \tau_b + \Delta t < t, \end{cases}$$ (10)

where $\Delta t$ is the signal's duration i.e., $\Delta t \sim \tau_s$, which is much smaller than $\tau_m$.

### Spiking density of LIF neuron after signaling input arrival.
Here, we derive the probability density of a postsynaptic spike occurring after arrival of the signaling input. To do so, we reset the time origin to the signal's arrival time in a way that the last postsynaptic spike happened at $\tau_b$ before the new origin. The conditional probability that the next postsynaptic spike happens at $\tau$ after the signal arrives is calculated as in Shomali et al.[36]

$$f_A(\tau|\tau_b) = \frac{J_A(\tau + \tau_b)}{1 - \int_0^{\tau_b} J_0(s)ds},$$ (11)

where the denominator is a normalization term to satisfy $\int_0^\infty f_A(\tau|\tau_b)d\tau = 1$. Next, we compute the probability density that the postsynaptic neuron generated a spike at $\tau_b$ before arrival of the signal, but not since then, $p_{back}(\tau_b)$. It is the probability of backward recurrence time, following renewal point process theory[104]: $p_{back}(\tau_b) = \mu(1 - \int_0^{\tau_b} J_0(s)\,ds)$, where $\mu = (\int_0^\infty sJ_0(s)ds)^{-1}$ is the mean firing rate of the postsynaptic neuron when there is no signaling input. By marginalizing Eq. (11) with respect to $\tau_b$ by using $p_{back}(\tau_b)$, we obtain[36]

$$f_A(\tau) = \int_0^\infty f_A(\tau|\tau_b) \times p_{back}(\tau_b)\,d\tau_b$$
$$= \mu \int_0^\infty J_A(\tau + \tau_b)\,d\tau_b.$$ (12)

Note that when the amplitude of the signaling input, $A$, is zero, $J_A(V_\theta, t) = J_0(V_\theta, t)$ and $f_A(\tau)$ simplify to $f_0(\tau) = \int_\tau^\infty \mu J_0(V_\theta, s)\,ds$.

Now, we can determine the probability of having one or more spikes in a specific time window, $\Delta$, after stimulus onset. It is given as the cumulative density function of $f_A(\tau)$:

$$F_A(\Delta) = \int_0^\Delta f_A(\tau)d\tau, \tag{13}$$

where the subscript $A$ indicates that $F_A(\Delta)$ is a function of the amplitude of the signaling input. In the absence of the signaling input (i.e., $A = 0$), we have $F_0(\Delta) = \int_0^\Delta f_0(\tau)d\tau$.

**Pairwise and triple-wise interactions of neural populations**. Using a binary representation of spiking activity for each postsynaptic neuron in a time window, $\Delta$ (schematically illustrated in Fig. 2a), we can represent the population activity of the postsynaptic neurons as a binary pattern. From the probabilities of the occurrence of all possible patterns, we can assess pairwise or higher-order interactions of the neural population. First, we consider two neurons. Let $x_i = \{0, 1\}$ ($i = 1, 2$) be a binary variable, where $x_i = 1$ means that the $i$th neuron emitted one or more spikes in the bin, while $x_i = 0$ means that the neuron was silent.

We denote by $P(x_1, x_2)$ the probability mass function of the binary activity patterns of the two postsynaptic neurons. Here, $P(1, 1)$ and $P(0, 0)$ are the probabilities that both neurons are, respectively, active and silent within $\Delta$. Similarly, $P(1, 0)$ is the probability that neuron 1 emits one or more spikes, while neuron 2 is silent during $\Delta$; $P(0, 1)$ represents the opposite situation. The probability mass function can be represented in the form of an exponential family distribution:

$$P(x_1, x_2) = \exp(\theta_1 x_1 + \theta_2 x_2 + \theta_{12} x_1 x_2 - \psi), \tag{14}$$

where $\theta_1$, $\theta_2$, and $\theta_{12}$ are canonical parameters, and $\psi$ is a log-normalization parameter. In particular, $\theta_{12}$ is an information-geometric measure of the pairwise interaction[37,60,63]. Accordingly, the pairwise interaction parameter is computed as

$$\theta_{12} = \log\frac{P(1, 1)P(0, 0)}{P(1, 0)P(0, 1)}. \tag{15}$$

We have $\theta_{12} = 0$ when the binary activities of two neurons are independent.

The same treatment can be applied to three neurons. The probability mass function for three neurons is written in exponential form as

$$P(x_1, x_2, x_3) = \exp\left(\sum_{i=1}^{3}\theta_i^t x_i + \sum_{i<j}\theta_{ij}^t x_i x_j + \theta_{123} x_1 x_2 x_3 - \psi\right). \tag{16}$$

Let us suppose $\theta_{123}$ (the triple-wise interaction parameter) is 0. In this case, the distribution reduces to the pairwise maximum entropy model, i.e., the least structured model that maximizes the entropy given that the event rates of individual neurons and joint event rates of two neurons are specified[105]. Consequently, a positive (negative) triple-wise interaction indicates that the three neurons generate synchronous events more (less) often than the chance level expected from the event rates of individual neurons and their pairwise correlations. From this equation, the triple-wise interaction among three neurons for the exponential family of probability mass function is calculated as[37,38]:

$$\theta_{123} = \log\frac{P(1, 1, 1)P(1, 0, 0)P(0, 1, 0)P(0, 0, 1)}{P(0, 0, 0)P(0, 1, 1)P(1, 0, 1)P(1, 1, 0)}. \tag{17}$$

**Mixture model of three neurons receiving common inputs to their pairs (triangle architecture)**. The mixture model of three neurons whose pairs receive independent common inputs (a triangle architecture) is calculated as follows. Figure 3d, shows eight possible patterns for the three independent common inputs. When the common input to neuron 1 and 2 is active (and the other two common inputs are silent), the pattern probabilities of three postsynaptic neurons are given by $P_A^1(\mathbf{x}) = [\prod_{i=1}^{2} F_A(\Delta)^{x_i}(1 - F_A(\Delta))^{1-x_i}] \times [F_0(\Delta)^{x_3}(1 - F_0(\Delta))^{1-x_3}]$. This situation happens in $(\lambda\Delta)(1-\lambda\Delta)^2 \times 100\%$ of the bins. The probabilities of activity patterns in which neurons receive the second (third) common input to neuron 2 and 3 (3 and 1), $P_A^2(\mathbf{x})$ ($P_A^3(\mathbf{x})$), obey equations similar to this one. The common inputs may be simultaneously applied to the same bin due to their independence. Namely, two common inputs coincide at $(\lambda\Delta)^2(1 - \lambda\Delta) \times 100\%$ of the bins. The pattern probability in the bins at which common inputs 1 and 2 coincide is given by $P_A^{12}(\mathbf{x}) = [\prod_{i=1,3} F_A(\Delta)^{x_i}(1 - F_A(\Delta))^{1-x_i}] \times [F_{2A}(\Delta)^{x_2}(1 - F_{2A}(\Delta))^{1-x_2}]$. Similarly, we define $P_A^{23}(\mathbf{x})$ and $P_A^{13}(\mathbf{x})$ for the bins at which the common inputs 2 and 3 and common inputs 1 and 3 coincide, respectively. Finally, all common inputs coincide at $(\lambda\Delta)^3 \times 100\%$ of the bins, for which the pattern probability is given by $P_A^{123}(\mathbf{x}) = \prod_{i=1,2,3} F_{2A}(\Delta)^{x_i}(1 - F_{2A}(\Delta))^{1-x_i}$. The parallel spike sequences are modeled as a mixture of these probability mass functions,

$$P(\mathbf{x}) = (1 - \lambda\Delta)^3 P_0(\mathbf{x}) + \sum_{i=1}^{3}(\lambda\Delta)(1 - \lambda\Delta)^2 P_A^i(\mathbf{x})$$
$$+ \sum_{i<j}(\lambda\Delta)^2(1 - \lambda\Delta)P_A^{ij}(\mathbf{x}) + (\lambda\Delta)^3 P_{2A}^{123}(\mathbf{x}). \tag{18}$$

For the asymmetric case, when two common inputs are shared among three neurons (Fig. 3e), the mixture model simplifies to

$$P(\mathbf{x}) = (1 - \lambda\Delta)^2 P_0(\mathbf{x}) + \sum_{i=1}^{2}(\lambda\Delta)(1 - \lambda\Delta)P_A^i(\mathbf{x}) + (\lambda\Delta)^2 P_A^{12}(\mathbf{x}). \tag{19}$$

**Sequential Bayesian estimation of neuronal interactions**. We extended the log-linear model for two (Eq. (14)) and three neurons (16) to a time-dependent model by using the time-dependent parameters collectively denoted as $\boldsymbol{\theta}_t$ and by assuming the following state transitions for the parameter:

$$\boldsymbol{\theta}_t = \boldsymbol{\theta}_{t-1} + \boldsymbol{\xi}_t, \tag{20}$$

where $\boldsymbol{\xi}_t$ is a Gaussian random variable with zero mean and covariance $\mathbf{Q}$. A sequential Bayes filter and smoother was performed to obtain the approximate posterior of the time-series $\boldsymbol{\theta}_1, \boldsymbol{\theta}_2, \ldots$ given the population activity data. The algorithm provides the mean and covariance of the approximated Gaussian posterior at every time step. The hyperparameter $\mathbf{Q}$ is optimized under the principle of maximizing marginal likelihood by using the expectation-maximization algorithm (see refs. [54,74] for the details of this method). We used a diagonal covariance matrix as $\mathbf{Q}$ whose entries for the first-order parameters were nonzero and optimized. The variances for the pairwise or higher-order interactions were set to zero, resulting in the estimation of the constant parameters.

**Ethics statement**. All procedures in Ohiorhenuan and Victor[58] were in accordance with the National Institutes of Health guidelines for the use and care of experimental animals and were approved by the Weill Cornell Medical College Institutional Animal Care and Use Committee. All experimental work for the Allen Brain Observatory—Neuropixels Visual Coding dataset was performed with approval and oversight of the Allen Institute Institutional Animal Care and Use Committee.

**Reporting summary**. Further information on research design is available in the Nature Portfolio Reporting Summary linked to this article.

## Data availability

The source data of this study is available at https://doi.org/10.5281/zenodo.7546537. The datasets analyzed in this study are from Ohiorhenuan and Victor[58] and the Allen Brain Observatory—Neuropixels Visual-Coding dataset, https://portal.brain-map.org/explore/circuits/visual-coding-neuropixels.

## Code availability

The custom codes of this study were written in MATLAB, Mathematica, Python, and NEURON. The codes are available at https://doi.org/10.5281/zenodo.7546537.

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

## Acknowledgements

S.R.S. acknowledges J. Victor's and C. Clopath's helpful comments; H.S. thanks S. Koyama and R. Kobayashi for valuable discussions and S.S. Hindupur Ravindra for providing code for preprocessing the Allen Brain Observatory data; S.R.S and S.N.R acknowledge T. Fukai's kind support and are grateful to him and H. Maboudi for valuable discussions. H.S. was supported by the National Institutes of Natural Sciences (NINS Program No. 01112005, 01112102), JSPS KAKENHI Grant Number JP 20K11709, 21H05246, and the Cooperative Intelligence Joint Research Program between Kyoto University and Honda Research Institute Japan.

## Author contributions

S.R.S., M.N.A., S.N.R., and H.S. designed the study. S.R.S. and H.S. performed the research. S.R.S., S.N.R., and H.S. analyzed the results. S.R.S., M.N.A., S.N.R., and H.S. wrote the first draft of the paper. S.R.S., S.N.R., and H.S. wrote and edited the paper.

## Competing interests

The authors declare no competing interests.
