## [Peer Review File · Communications Biology]

Reviewers' comments:

Reviewer #1 (Remarks to the Author):

Review for "Uncovering Network Architecture Using an Exact Statistical 1 Input-Output Relation of a Neuron Model"

The paper tries to supply a framework for uncovering the shared input connectivity of neurons recorded in experimental settings (In-vivo or in-vitro). The paper is well written, well explained, and well-executed. The main issue with the work is that it is not tested or verified, indeed it is hard to get experimental exact detail on shared input to verify the method, but the method should at least pass testing on a detailed model of a multi-compartmental network of 3 neurons.

I understand that the skillset needed to run multi-compartmental modeling might be different from the one the authors have, but the testing required will not take a significant amount of time from a researcher that already has this skill set.

The reviewers did show that their method is valid both for LIF and Adaptive LIF, but as it is proposing a map for interpretation of experimental results (Fig12), the threshold is higher. Pyramidal neurons in V1 of a mouse (or monkey/human) have many different ion channels, additionally, they have both NMDA & AMPA synapses that cause non-linear dendritic interaction, there is also dendritic attenuation and synaptic failure. These biophysical mechanisms will add a lot of noise, and it is needed to show that this biological variability does not change the analytic single compartment results.

Having access to a computer cluster, it is possible, in a reasonable amount of time, to run Fig 3,6,7 on different multi-compartmental models of V1 pyramidal cells and verify that the prediction hold.

Additionally (and at the bare minimum), they should simulate a collection of such 3-neurons networks, with different input motifs and verify that indeed the output θ_{123} versus θ_{12} fits the network connectivity that was set in the simulation.

There are many repositories of cells that can be used to build such a network, 2 that I can suggest are the Blue-Brain Project of Allen institution:

<http://celltypes.brain-map.org/data>

<https://bbp.epfl.ch/nmc-portal/welcome.html>

(I would suggest using a mouse model and not a monkey, as mouse single neuron models are more advanced).

Other comments:

1. It will be beneficial to supply a bit more details about the simulations, maybe also a chart for all the variables (time constants, simulation time, synaptic conductance etc..) for each simulation for each figure.

2. Can you supply some intuition on why in Figure 3A the Simulation results are ~ 1 and the theory is 0.8 for the case of Bin width of 3ms?

3. Can you verify the results in Figure 3B by running stimulations (single compartment, as in Fig 3A) that will show that the trend is correct, I understand that running the full matrix of options might be computationally heavy, but it will help to verify a few trend lines across the matrix.

4. "A single common excitatory input, in the star architecture, significantly increases the probability that all three neurons spike in the observation time window of Δ , $P(1, 1, 1)$, whereas a single common in-hibitory input increases the probability of the reverse pattern, $P(0, 0, 0)$. This simply changes the sign of θ in Eq. 17 (Methods)."

This statement is not shown in figure 5C, as it is not only that the value of θ is reversed it is also decreased. (Max -0.06 vs 2), can you please discuss it and fix the statement.

Also for Figure 5C Triangular architecture (Maybe it would be better to mark it with D), the value of θ is 0.001 compared to the excitatory case of 0.4

5. "Each motif's boundaries in the plane of θ_{123} versus θ_{12} are shown in Figure 7B. ", Figure 7 does

not have labels.

6. In the legend of Fig. 8 – “whereas the orange lines show the wide region for excitatory inputs given to three neurons” – there is no orange line in the figure. I assume you mean green?

7. It might be better to mark the areas in Fig8 and not just draw the lines (e.g.

https://matplotlib.org/stable/gallery/lines_bars_and_markers/line_between_demo.html)

8. Missing details about language that was used (Matlab, Python, R, C ?)

9. The amount of recording time needed to get the correct results is not stated, it will be beneficial to show how the results change with recording time (assuming infinite time will result with the analytic results)

10. A simple Python/Matlab script that calculates both θ_{123} versus θ_{12} would increase the use of the suggested method, I advise of supplying one.

Reviewer #2 (Remarks to the Author):

Summary:

The authors present an analytical framework for inferring local input architecture to neural trios based on higher-order (pair- and triple-wise) interaction terms in an information geometric measure applied to spiking statistics. Moreover, they use their framework to predict the most likely local connectivity scheme giving rise to measured spiking activity in anesthetized primate (macaque) V1. They find that the strong negative triple-wise (but positive pair-wise) interactions observed in the original recordings are best explained by excitatory inputs to pairs of neurons in each trio, rather than by mutually shared inhibition (the intuitive explanation). Furthermore, they probe the robustness of their results with respect to various assumptions including: 1) the model of neural spiking underlying their analytical framework, 2) the activity balance regime (near-threshold, subthreshold, superthreshold), 3) the presence of reciprocal/directional connections between neurons within the trio. In all cases, they find that interaction strengths are bounded by input amplitude on one side and postsynaptic spontaneous activity rates on the other. Finally, they compare the ability of their framework to distinguish among possible input architectures against more conventional methods (including pair-wise cross-correlation and pair-wise covariance), finding that their proposed framework consistently outperforms the alternatives. Based on their analysis, they provide an interaction strength “guide map” to researchers who wish to replicate their approach.

What follows below are general comments regarding content and language/style. However, the bulk of this review is contained in the annotated pdf attached. Please refer to this document for detailed questions and suggestions.

General comments (content):

This paper was a pleasure to read, and certainly merits publication and dissemination to the community. I found the arguments thorough and well-reasoned with few exceptions. Furthermore, care was taken to clearly enumerate the assumptions and potential shortcomings of the approach, and to address each systematically. Also, I find the proof-of-principle intriguing and potentially useful for future research, pending additional validation and vetting on experimental datasets (with ground-truth connectivity data).

However, I do have a few nagging questions:

First: what if the animal were not anesthetized? Would we still see strong negative triple-wise interactions? I wonder if there’s an easy way to test this empirically (rather than relying on theoretical arguments). Does Ohiorhenuan have any awake data? Or perhaps there are other publicly available datasets that would meet the requirements for such a test?

Second: it wasn't clear to me that the guide map would be capable of discriminating among input architectures for all cases. For example, what if you needed to distinguish between asymmetric excitatory inputs to pairs and excitatory inputs to trios for low spontaneous firing rates? Or equivalently, asymmetric inhibitory inputs to pairs and inhibitory inputs to trios for high spontaneous rates?

Third: unless I missed something, the conclusion that excitatory inputs to pairs best explains the data appears to be strictly qualitative (i.e. it fits best "by eye"). Could you include a quantification of the "goodness of fit" (with uncertainty) for each of the possible architectures to put your conclusion on numerical grounds?

Fourth: there is a bit of a fundamental gap, in that there is no ground truth to test against. Can you please expand your comments on this issue (lines 677-680), suggesting specific future experiments to address the problem?

General comments (style and language):

In the annotated pdf attached, I've made an attempt to fix the most egregious grammatical errors and awkward phraseology. However, these corrections are not sufficient to bring the language across the "acceptability threshold" for publication. I strongly urge you to pass the manuscript either to a colleague or professional copyeditor for a thorough reworking.

As a stylistic point, the entire text is written in present tense, even descriptions of specific experiments presumably carried out in the past. This is distracting since it causes many statements to feel inappropriately general. I recommend using the present tense for discussing general principles and other temporally non-specific topics. For describing temporally-constrained events, please use the appropriate tense. For example, OLD: "To improve the biological plausibility of our neuron model, we add an adaptation term to the LIF neurons and run simulations to investigate how this influences pair- and triple-wise interaction strengths."

NEW: "To improve the biological plausibility of our neuron model, we added an adaptation term to the LIF neurons and ran simulations to investigate how this would influence pair- and triple-wise interaction strengths."

Many words and phrases are italicized in a seemingly arbitrary fashion. I would recommend avoiding all italicization except when used to emphasize a point (usually in opposition to another, or against naive expectation).

The names of the various input architectures (excitatory-to-trio etc.) need revision as they are too cumbersome. Please choose a simple naming scheme (perhaps using acronyms?) and apply it consistently.

Many figure captions contain interpretive text. Please refrain from adding any text that is not purely descriptive of the relevant figure in the captions.

Reviewers' comments: black

Our responses: blue

Reviewer #1 (Remarks to the Author):

Review for “Uncovering Network Architecture Using an Exact Statistical 1 Input-Output Relation of a Neuron Model”

The paper tries to supply a framework for uncovering the shared input connectivity of neurons recorded in experimental settings (In-vivo or in-vitro). The paper is well written, well explained, and well-executed. The main issue with the work is that it is not tested or verified, indeed it is hard to get experimental exact detail on shared input to verify the method, but the method should at least pass testing on a detailed model of a multi-compartmental network of 3 neurons. I understand that the skillset needed to run multi-compartmental modeling might be different from the one the authors have, but the testing required will not take a significant amount of time from a researcher that already has this skill set.

The reviewers did show that their method is valid both for LIF and Adaptive LIF, but as it is proposing a map for interpretation of experimental results (Fig12), the threshold is higher. Pyramidal neurons in V1 of a mouse (or monkey/human) have many different ion channels, additionally, they have both NMDA & AMPA synapses that cause non-linear dendritic interaction, there is also dendritic attenuation and synaptic failure. These biophysical mechanisms will add a lot of noise, and it is needed to show that this biological variability does not change the analytic single compartment results.

Having access to a computer cluster, it is possible, in a reasonable amount of time, to run Fig 3,6,7 on different multi-compartmental models of V1 pyramidal cells and verify that the prediction hold.

Additionally (and at the bare minimum), they should simulate a collection of such 3-neurons networks, with different input motifs and verify that indeed the output θ_{123} versus θ_{12} fits the network connectivity that was set in the simulation.

There are many repositories of cells that can be used to build such a network, 2 that I can suggest are the Blue-Brain Project of Allen institution:

<http://celltypes.brain-map.org/data>

<https://bbp.epfl.ch/nmc-portal/welcome.html>

(I would suggest using a mouse model and not a monkey, as mouse single neuron models are more advanced).

Thank you very much. According to the reviewer's suggestion, we performed the simulation of the multi-compartmental model of pyramidal neurons in Layer 5 using the resources from the

Blue Brain Project (<https://bbp.epfl.ch/nmc-portal>, L5_TTPC1_cADpyr232_1). The result is presented in Figure 6B and the details come at Supporting Information V.

These neurons possess multiple m-type excitatory and inhibitory synapses at different layers of L2-3, L4, L5, and L6. These m-type synaptic inputs are Poissonian events of the frequency 10 Hz with their synaptic dynamics and morphology. On top of all these synapses, we introduced a Poissonian presynaptic neuron conveying our shared signaling input with specific frequency and amplitude (details in Supporting Information V). We simulated the activity of 3 postsynaptic neurons that receive the same time series of a signaling input as shared input (for the star-architecture). We repeated this 5 times each with a different Poissonian shared input, to obtain the standard deviation of the interactions. Note that other synaptic events are random and independent across the trials. This procedure could simulate the postsynaptic neurons receiving common (excitatory or inhibitory) inputs. We performed the simulation of 3 neurons while changing the amplitude of the shared signaling input. We took a similar approach to simulate the triangular structure (excitatory or inhibitory-to-pairs of neurons, Supporting Information V).

We plotted the θ_{123} versus θ_{12} obtained from this biophysical simulation in Figure 6B, together with theoretical boundaries for each motif. The result confirms that our theoretical boundaries predict the architecture behind the activities of the network of three multicompartmental model neurons.

Other comments:

1. It will be beneficial to supply a bit more details about the simulations, maybe also a chart for all the variables (time constants, simulation time, synaptic conductance etc..) for each simulation for each figure.

We agree with the reviewer's suggestion. We consider that we would need such tables for the adaptive neuron model because it is a complicated model and includes all parameters for simulating a LIF neuron model. We added the table of the used variables (Table.1, Supporting Information VII). The LIF model without adaptive term is included in this table. In addition, we note that the legend of each figure contains information required to simulate the simple LIF neuron model. The parameters of the multicompartmental model follow the blue brain project. We provided the relevant information on the neuron we used in the Supplementary Information V.

2. Can you supply some intuition on why in Figure 3A the Simulation results are ~ 1 and the theory is 0.8 for the case of Bin width of 3ms?

Thank you for raising this question. This discrepancy is caused by the violation of assumption for the mixture model when the very small bin (1 ms) is used. In order to more specifically point out the limitation of our model, we brought the description of the carry-over effect that is not

included in the mixture model right after when we introduced the mixture model (Eq.1) and introduced the new Supporting Information I about this effect. We also stated that there is a discrepancy between the theory and simulation for the small bin size.

3. Can you verify the results in Figure 3B by running stimulations (single compartment, as in Fig 3A) that will show that the trend is correct, I understand that running the full matrix of options might be computationally heavy, but it will help to verify a few trend lines across the matrix.

We picked up the parameters corresponding to several points in Fig. 2C (previous 3B) and ran the simulation. This result was presented as Supporting Figure S2 in Supporting Information I. We stimulated LIF neurons with shared input on top of Gaussian noisy inputs for 10^8 steps with step size 0.1 ms and repeated it for 500 trials. Supporting Figure S2 Left, shows the pairwise interaction of 2 neurons for different values of scaled amplitudes while fixing the scaled diffusion for inhibitory and excitatory common inputs. This figure shows that, in both inhibitory and excitatory common input cases, as signaling input gets stronger, the pairwise interaction increases and then saturates as the theory (Fig.2C) suggests. So "the trend is correct". We also checked the trend for triple-wise interactions for some values of scaled diffusion and scaled amplitude of signaling inputs. The result shows the same trend we found by analytics. The slight discrepancy between the theory and simulation is explained by the carry-over effect that we did not assume for the mixture model of the population activity. In the revised manuscript, we added Supporting Information I that explains this effect.

4. "A single common excitatory input, in the star architecture, significantly increases the probability that all three neurons spike in the observation time window of Δ , $P(1, 1, 1)$, whereas a single common in-hibitory input increases the probability of the reverse pattern, $P(0, 0, 0)$. This simply changes the sign of θ in Eq. 17 (Methods)."

This statement is not shown in figure 5C, as it is not only that the value of θ is reversed it is also decreased. (Max -0.06 vs 2), can you please discuss it and fix the statement.

Also for Figure 5C Triangular architecture (Maybe it would be better to mark it with D), the value of θ is 0.001 compared to the excitatory case of 0.4

When the postsynaptic neuron's firing rate is low, the effect of inhibitory common inputs to postsynaptic neurons is weak, regardless of the architecture. We mentioned this fact and its reason in greater detail in the section titled "Excitation versus inhibition: Which one can produce stronger triple-wise interactions?".

According to the reviewer's point, we remarked this issue, and stated that this will be discussed in the later section in the end of this paragraph as follows:

"We also observed that the magnitudes of the negative triplewise interactions induced by inhibitory common inputs are much weaker than the excitatory common inputs. We expect this effect when the postsynaptic neuron exhibits low spontaneous firing rate. In the subsequent section, "Excitation versus inhibition: Which one can produce stronger triple-wise interactions?"

and in Fig~\ref{Whyinhcan't}, Supporting Information III, we discussed why inhibitory inputs cannot generate strong interactions in low spontaneous firing rates."

5. "Each motif's boundaries in the plane of θ_{123} versus θ_{12} are shown in Figure 7B. ", Figure 7 does not have labels.

Thank you. We corrected Figure 7B to Figure 7A right panel. The new version of Figure 7 has panel B which is a multicompartmental simulation result.

6. In the legend of Fig. 8 – "whereas the orange lines show the wide region for excitatory inputs given to three neurons " – there is no orange line in the figure. I assume you mean green?

Thank you very much. We corrected it to 'green'.

7. It might be better to mark the areas in Fig8 and not just draw the lines (e.g. https://matplotlib.org/stable/gallery/lines_bars_and_markers/line_between_demo.html)

Thank you for the suggestion. Indeed, we tried such a colored map, but found it difficult to construct a clear picture since Fig. 7 (previous 8) is already crowded. Therefore, we wish to leave it as it is.

8. Missing details about language that was used (Matlab, Python, R, C ?)

We provided the code information in the "Code availability" section. Additionally, the same information will be included in Summary Checklist requested by Springer-Nature.

9. The amount of recording time needed to get the correct results is not stated, it will be beneficial to show how the results change with recording time (assuming infinite time will result with the analytic results)

We provided the dependency of the estimated interactions on the simulation length for the multi-compartmental model (Fig.S7), and confirmed that we used the adequate simulation length to obtain the converged values.

10. A simple Python/Matlab script that calculates both θ_{123} versus θ_{12} would increase the use of the suggested method, I advise of supplying one.

We are grateful for the advice. We will provide the code at the time of publication, together with other codes used in the paper.

Reviewer #2 (Remarks to the Author):

Summary:

The authors present an analytical framework for inferring local input architecture to neural trios based on higher-order (pair- and triple-wise) interaction terms in an information geometric measure applied to spiking statistics. Moreover, they use their framework to predict the most likely local connectivity scheme giving rise to measured spiking activity in anesthetized primate (macaque) V1. They find that the strong negative triple-wise (but positive pair-wise) interactions observed in the original recordings are best explained by excitatory inputs to pairs of neurons in each trio, rather than by mutually shared inhibition (the intuitive explanation). Furthermore, they probe the robustness of their results with respect to various assumptions including: 1) the model of neural spiking underlying their analytical framework, 2) the activity balance regime (near-threshold, subthreshold, superthreshold), 3) the presence of reciprocal/directional connections between neurons within the trio. In all cases, they find that interaction strengths are bounded by input amplitude on one side and postsynaptic spontaneous activity rates on the other. Finally, they compare the ability of their framework to distinguish among possible input architectures against more conventional methods (including pair-wise cross-correlation and pair-wise covariance), finding that their proposed framework consistently outperforms the alternatives. Based on their analysis, they provide an interaction strength "guide map" to researchers who wish to replicate their approach.

What follows below are general comments regarding content and language/style. However, the bulk of this review is contained in the annotated pdf attached. Please refer to this document for detailed questions and suggestions.

Thank you so much for your valuable comments and detailed suggestions in this letter and the PDF. We applied changes to the main text following these suggestions. Below we provide our responses to the general comments, which are followed by all of the comments in the PDF and our replies to them.

General comments (content):

This paper was a pleasure to read, and certainly merits publication and dissemination to the community. I found the arguments thorough and well-reasoned with few exceptions. Furthermore, care was taken to clearly enumerate the assumptions and potential shortcomings of the approach, and to address each systematically. Also, I find the proof-of-principle intriguing and potentially useful for future research, pending additional validation and vetting on experimental datasets (with ground-truth connectivity data).

However, I do have a few nagging questions:

First: what if the animal were not anesthetized? Would we still see strong negative triple-wise interactions? I wonder if there's an easy way to test this empirically (rather than relying on

theoretical arguments). Does Ohiorhenuan have any awake data? Or perhaps there are other publicly available datasets that would meet the requirements for such a test?

Thank you for asking this question. We agree that it is valuable to test our method for awake animals and perform the analysis of higher-order interactions by ourselves. We thus performed additional data analysis using the Allen Institute's visual coding data sets containing spiking neural activity recorded by neuropixels probes (<https://portal.brain-map.org/explore/circuits/visual-coding-neuropixels>). We picked up neurons in the primary visual cortex responding to drifting grating stimuli (the same as for the monkeys) as a prototypical experimental setup of neurophysiology. In conclusion, we found the same positive pairwise and negative triple-wise interactions that should be explained by the excitatory-to-pairs motif. This result was summarised in Figure 8 in the new manuscript.

One issue to be resolved in dealing with the awake behaving animals is to account for the time-varying firing rates of individual neurons. Estimating neural correlations while ignoring the time-varying firing rates can result in pseudo correlations. Previously, we developed the state-space analysis of higher-order interactions (Ref. 1 below). We added the brief description on this method in Methods section. This method performs a sequential Bayesian filtering and smoothing estimation of the time-varying firing rates and neuronal interactions with the Bayesian credible intervals while optimizing its hyper-parameters, including the one that tunes the time-scale of their dynamics. Thus this analysis also replies to questions raised in other places regarding the quantification of uncertainty in the data. Using this method, we estimated the pairwise and third-order interactions while estimating the time-varying firing rates of individual neurons (Fig.8B). The analyses of V1 neurons in multiple stimuli and mice from this data set corroborate that the V1 neurons express significant positive pairwise and negative triple-wise interactions, suggesting locally excitatory common inputs to pairs as an underlying architecture (Fig.8E). We also confirmed this result by repeating our analysis using the trial-shuffled data. Neurons in this data show the same time-varying firing rates and do not contain significant interactions beyond sampling fluctuations, indicating that the result is not an artifact of time-varying rates.

1. Shimazaki H., Amari S., Brown E. N., and Gruen S., State-space Analysis of Time-varying Higher-order Spike Correlation for Multiple Neural Spike Train Data. PLoS Comput Biol (2012) 8(3): e1002385. doi:10.1371/journal.pcbi.1002385

Second: it wasn't clear to me that the guide map would be capable of discriminating among input architectures for all cases. For example, what if you needed to distinguish between asymmetric excitatory inputs to pairs and excitatory inputs to trios for low spontaneous firing rates? Or equivalently, asymmetric inhibitory inputs to pairs and inhibitory inputs to trios for high spontaneous rates?

We agree that, since some of the boundaries are close, it may be challenging to decide the underlying architecture for some real data. As you pointed out in the third comment, it is important to provide confidence bounds for the estimated interactions to decide whether the variability in the data allows for making clear distinctions or not. Let us address this point in the next comment.

Third: unless I missed something, the conclusion that excitatory inputs to pairs best explains the data appears to be strictly qualitative (i.e. it fits best "by eye"). Could you include a quantification of the "goodness of fit" (with uncertainty) for each of the possible architectures to put your conclusion on numerical grounds?

We thank the reviewer for raising this important question on the accuracy of judgment when using our guide map. First, we remark that, to obtain the goodness-of-fit and perform model comparison, we need to construct statistical models (such as GLM) of neurons that can be fitted to data using standard fitting algorithms. Such a model would receive common hidden inputs as latent variables. While such statistical model allows us to compute the "goodness of fit" of the model, such models are relatively simpler and physiologically implausible compared to the mechanistic models of neuron (such as LIF), and limited to specific choices of the model architecture.

Instead, in this article, we provided the analytic boundaries of the population statistics (pairwise and higher-order interactions) for LIF and extended models by covering physiological ranges of their parameters. We obtained these exact analytic predictions by taking an expectation over the probabilities: it is an average over infinite possible samples/trials of an ensemble. This approach allows us to use the data without fitting a model to data. In this approach, instead of the model goodness-of-fit, we need to assess the uncertainty of the interactions estimated from finite samples of experimental/simulation processes.

As for Ohiorhenuan and Victor's result, we do not have any access to their accuracy (or, conversely, error bar) for each data point in their paper. We thus performed additional data analysis of higher-order interactions from awake mice in the revised manuscript partly in response to the previous question (Fig.8). The Bayesian method used there provides credible intervals for the estimated interaction. We used them to select groups of neurons showing significant pairwise and triple-wise interactions. Further, the results show that the credible intervals of most groups are within the theoretical boundaries explained by the excitatory-to-pairs motif. We thus demonstrated that providing the confidence bound for data is critical to judge the underlying architecture.

Following the reviewer's constructive concern, we added a sentence to recommend considering 'confidence bound' when judging the architecture when we introduced the guide map in the revised manuscript:

"It is recommended to consider uncertainty in estimating the empirical interactions when judging the architecture to avoid erroneous detection of the architecture."

General comments (style and language):

In the annotated pdf attached, I've made an attempt to fix the most egregious grammatical errors and awkward phraseology. However, these corrections are not sufficient to bring the language across the "acceptability threshold" for publication. I strongly urge you to pass the manuscript either to a colleague or professional copyeditor for a thorough reworking.

We sincerely appreciate the tremendous effort in correcting our non-native English by the reviewer. We certainly did not expect such a kind offer from the reviewer, and truly thankful for that. We changed our manuscript following the suggestions of the reviewer. Although the manuscript is not yet sent out for English proofreading, we are ready to send it before the publication, following the editor and reviewer's recommendation.

As a stylistic point, the entire text is written in present tense, even descriptions of specific experiments presumably carried out in the past. This is distracting since it causes many statements to feel inappropriately general. I recommend using the present tense for discussing general principles and other temporally non-specific topics. For describing temporally-constrained events, please use the appropriate tense. For example, OLD: "To improve the biological plausibility of our neuron model, we add an adaptation term to the LIF neurons and run simulations to investigate how this influences pair- and triple-wise interaction strengths."

NEW: "To improve the biological plausibility of our neuron model, we added an adaptation term to the LIF neurons and ran simulations to investigate how this would influence pair- and triple-wise interaction strengths."

Following the reviewer's suggestion, we used the past tense for the procedures we performed in the past. We kept the present tense for the description of the mathematical results.

Many words and phrases are italicized in a seemingly arbitrary fashion. I would recommend avoiding all italicization except when used to emphasize a point (usually in opposition to another, or against naive expectation).

Thank you so much. In the revised manuscript, we significantly reduced the use of italic styles.

The names of the various input architectures (excitatory-to-trio etc.) need revision as they are too cumbersome. Please choose a simple naming scheme (perhaps using acronyms?) and apply it consistently.

We went over the manuscript and consistently used the architecture names such as excitatory-to-pairs or inhibitory-to-trio. We consider that these are the simplest and intuitive naming scheme, and wish to avoid using the acronyms.

Many figure captions contain interpretive text. Please refrain from adding any text that is not purely descriptive of the relevant figure in the captions.

We removed all interpretive texts from the figure captions.

Response to the reviewers comments in PDF

Below we provide our point-by-point replies to the reviewer's comments made in the PDF files. The line numbers correspond to the ones in the previous submission. We've applied changes to the main text, responding to almost all of the comments directly written in the PDF. If not, we explain why we kept the current style of the present manuscript. There are also many suggestions for correcting English, which we incorporated into the revised manuscript.

L16 I somewhat object to this language, and also find it ambiguous. TBD later.

The first sentence of the abstract was rewritten. It now reads as:

"Identifying network architecture from observed neural activities is crucial in neuroscience studies."

L24 This can't really be intuitive without a picture of the "negative triple-wise interactions".

intuitieve was replaced with alternative.

L26 Too vague.

We rewrote the sentence as "We present a guide map of neural interactions to specify the hidden motifs underlying observed interactions found in empirical data."

L31 The work of Markram et al. above is not merely a structural connectome, but a functional predictive model informed by paired in vivo recordings.

We moved Markram et al. to the reference of the next sentence.

L53 Why would it? if moderate synaptic inputs can cause spiking, surely strong inputs could also?

We corrected the flow of logic. This part now reads as

"In this condition, even a moderate synaptic input can cause a spike in the postsynaptic neuron. However, given that the distribution of the synaptic strengths in cortical and hippocampal neurons follows the log-normal distribution \cite{buzsaki2014log}, it was suggested that fewer strong synaptic connections constitute the backbone of the microcircuit with the aid of inputs from the large number of weak synapses~\cite[{}]{lefort2009excitatory,song2005highly,cossell2015functional}."

L127 <formatting>

We checked the grammar and found that we should use the capital after the colon ':'. We thus wish to keep this style. However, we would be happy to change it if it is pointed out by English proofreading, again.

L132 comprising

Changed.

L134 missing a space between constant and units

A space was added.

L169 This is an exact duplication of the paragraph above.

Thank you. We removed it.

L185 I assume that λ must be defined such that it has a maximum value of 1 spike per bin duration (even if more spiking actually occurs)? Perhaps it would be good to clarify.

Yes, we added a sentence of clarification:

"Here we assume sparse signaling inputs so that the probability of more than one signaling inputs in a single bin, is small."

Fig 1 what is meant by "rest"? don't you mean "spread", or "diffusivity" or something similar?

Sorry, it was a typo. Removed.

L196 I don't understand this acronym (perhaps it's defined in the references?). What does it stand for? Please define it explicitly. Are you referring to the supplemental materials?

It stands for Supporting Information. To be clear, we use "Supporting Information" every time we refer to it.

L213 Saying that it "agrees" is non-specific, and probably belongs in the discussion rather than the results.

We rephrased the sentence more precisely:

"The pairwise interaction predicted by the mixture model are placed within the error bars of the simulation results except for the smallest bin, 1 ms"

L219 Yes, but primarily because any number of spikes in a bin is counted as either a 1 (some spikes happened) or a 0 (no spikes), correct? Maybe it's good to emphasize that the dependence on bin size is a consequence of this, otherwise it's counter-intuitive that distinguishing common inputs should be impossible for large bin size.

Yes, we agree with the reviewer's view. To clarify, this is caused by the binarization, we added ",with the binary representation," in the sentence.

L235 Looking forward to the discussion.

The sentence "In contrast to the common excitatory input case, pairwise interaction and $D/(\tau_m V_{\theta 2})$ are positively correlated" required further explanation. We made it more accessible by replacing it with:

"In contrast to the common excitatory input, where the effect of the common input is stronger for the low scaled diffusion coefficient (low firing rate), we observe stronger effect of the inhibitory common input under the higher scaled diffusion coefficient (high firing rate). This observation is further discussed in the section "Excitation versus inhibition: Which one can produce stronger triple- wise interactions?"

L314 what do you mean by "homogenous"?

By "homogeneous", we mean that the LIF model uses the same parameters. To be clear, we removed this term, and added a definite article, "the", to refer to the LIF neurons we used under this condition.

Fig.6 What about the effects of the bin size here?

We add a figure in Supporting Information IV (Fig S5) about how bin size can affect the boundaries in the plane of pairwise and triple-wise interactions.

Fig.6 The triangle symbols for "inhibitory inputs to pairs" look like arrows pointing backwards, making the interpretation of the direction of the scaled amplitude parameter confusing. Consider clarifying that the `_black_` arrows indicate the directionality.

Thank you for this point. Following the suggestion, we explicitly mentioned the black arrows in the caption of Figure 5 (previous 6).

Fig. 6 This is interpretation and does not belong in a figure caption.

Removed.

Fig.7 Interpretation does not belong in the captions.

Removed.

Fig.7 The color/linestyle scheme is very confusing and needs to be changed. Maybe it's confusing to call $u=1\text{Hz}$ and $u=100\text{Hz}$ analytic boundaries, if they are actually just de facto boundaries? Please reconsider this plot for clarity.

While we keep the figure style, we added more explanations on the boundaries and the lines for intermediate diffusions in the figure caption.

Fig. 7 These aren't visible. Somehow this figure needs to be reworked to be more clear. Also please explicitly mention that the reader should refer to figure 6 for the coloring scheme.

This figure was revised to include the results of the multicompartamental model, which additionally provides a picture magnified at the origin. We hope that the reviewer's concern is addressed by this revision. In addition, we added reference to Figure 6 for the color code.

L350 The use of italics throughout is odd and seemingly arbitrary.

We removed italics here. We also significantly reduced the use of italic words.

L359 weak

Removed italic style.

L380 Can you justify explicitly why the data are NOT consistent with the "inhibitory-to-trio" motif? I see that excitatory to pairs *_looks_* better, but can you rule it out with an uncertainty analysis?

We agree that the inhibitory-to-trio might explain one of the data samples from nearby neurons. To be precise, we rephrased the sentence as:

"In contrast, neither excitatory-to-trio nor any of the inhibitory motifs can explain most of the interactions of neurons within $300\ \mu\text{m}$."

We ruled out that the shared inhibition is behind the observed 'strong' negative triple-wise interactions. However, we do not rule out that other architectures, including the inhibitory-to-trio, are underlying the data sets showing weak interactions. For those groups showing weak interactions, we can not address the architecture nor differentiate them from the sample noise of neurons (neurons without any shared connections). For this reason, we improved several sentences in the manuscript to ensure that ruling out the inhibitory-to-trio as the underlying motif applies to neurons showing 'strong' interactions.

The uncertainty analysis requires data sets. We added the new analysis on mouse V1 neurons from Allen data sets, where we evaluated the uncertainty of the estimated interactions (Fig. 9). The results with the Bayesian credible intervals of the estimated interactions show that none of the groups exhibiting significant interactions fell on the regime of the inhibitory-to-trio architecture. Thus we ruled out the inhibitory-to-trio with an uncertainty analysis for this data set.

Fig.8 What is the difference between red triangles, red stars, and red squares?

We added "The triangles are from three neuron data sets. The star and square shapes are from the four- and five-neuron data sets."

Fig.8 What orange lines? Do you mean the green lines?

Yes it is green. Changed. Thank you.

Fig.8 Interpretation. Doesn't belong in caption.

Removed.

L408 Since you're describing an *_increasing_* relationship w.r.t. absolute value in the text, maybe you should flip the y-axis in Fig. 9 to make the trend more visually intuitive.

Since it is important to show that the negative triple-wise interactions are induced, we refrain from flipping the y-axis. We also removed the 'absolute value' in the main text.

L412 Yes, qualitatively. But I'm still not convinced from a numerical perspective. Can you quantify your certainty that inhibitory-to-trio is not a better fit?

We do not deny that the small negative triplewise interactions are also explained by the inhibitory-to-trio. To be precise, we added the following sentence:

"While the motifs 2-7 in the first cluster (blue) may not be distinguished from the inhibitory-to-trio (the motif 1), the strong negative triple-wise interactions observed in the second, third, and fourth cluster (the motifs 8-16) cannot be explained by the inhibitory motifs (the motif 1) even if the amplitude of the inhibition is increased (Fig~\ref{condthreeneurons})."

Regarding the quantification of uncertainty, please refer to our new analysis of the mouse V1 neurons and responses to L380 and Third question in the general comments.

L419 italicization?

Removed.

L424 citation?

Since this sentence may not be accurate given the sentence that follows (see below), we removed this sentence.

L425 This is too vague. Can you be more specific about the limitations of LIF models? Just a few words would be enough.

The sentence was revised as

"Although the LIF neuron model is the standard reduced model, it has some limitations, e.g., in reproducing variability of the inter-spike intervals observed in vivo"

Fig.9 excitatory-to-pairs

We decided to use excitatory-to-pairs consistently in the revised manuscript.

Fig.9 interpretation doesn't belong in caption

Removed.

Fig.10 What is this light brown dashed-dotted line?

This was an irrelevant line in this figure. We removed it. Thank you for pointing this out.

L434 Ok, Fig. S5 shows this, but it is not possible to see in Fig. 10.

It is exactly for this reason that we provided the supplementary figure. We simply added "for the details, see" to introduce the Fig. S5 in the previous version.

L439 strange choice of italicization - I would put it on *_strengthens_*, not adaptation.

To reduce the italics, we removed it from "adaptation".

L449 I don't see this. They look different to me in figure 10A?

We added further explanation for each model as

"aLIF (superposition of solid red and dashed light red lines) and aEIF (superposition of solid green and dashed light green lines)"

L467 Surely there are many, many more studies supporting the notion that cortex typically operates at a balance of EXC and INH near threshold???

Here we cited the study by Tan et al. that shows that the balanced conditions particularly appear during the stimulus presentation only because we analyze neuronal activity during the stimulus presentation. To be precise, we change the sentence as follows.

"the near-threshold assumption during the stimulus processing is partly supported by an empirical study~\cite[{}]{tan2014sensory}"

L474 Please re-word this sentence.

We rewrote the sentence as

"We remarked evidences that cortical neurons operate in the $\{\text{subthreshold}\}$ (or near the threshold) regime \cite[{}]{shadlen1998variable}, depending on the state of the animal or stimulus conditions \cite[{}]{tan2014sensory}, rather than the suprathreshold regime that results in regular spikings. "

L481 Still not convinced. Show me the analysis.

In the lines above L481, we explained the results of the analysis on the subthreshold regimes. We showed that making the equilibrium subthreshold does not change the interactions such that the architectures are not distinguishable. To make this point clear, we added the following sentence in the Overview of the Discussion section:

"These results obtained under the adaptation and subthreshold regimes does not oppose our original conclusion that the observed strong negative triple-wise interactions are signatures of excitatory-to-pairs."

Fig. 12 Where is the $\mu=100\text{Hz}$ data (dashed-dotted gray line)? I don't see it in the figure.

This was a mistake. We removed this label.

Fig. 12 Where do I see this in Fig. 12A? Are you referring to Fig. 12B?

This sentence was a typo. We removed it.

L505 The use of $F_0(\Delta)$ vs μ to refer to "spontaneous activity" seems pretty loose throughout. Can you do a consistency check? I guess one is a probability and one is a rate.

Yes, the former is the probability and the latter is the rate. In the revised manuscript, wherever we mention $F_0(\Delta)$, we consistently used "the spontaneous spiking probability". For μ , we used "spontaneous rate".

L528 I would comment here that in certain cases (such as Ohiorhenuan et al. 2010), your method works well. However, it would also seem that there are other scenarios in which determining the underlying neural architecture might be confounded. For example, what if you needed to distinguish between asymmetric excitatory inputs to pairs and excitatory input to trio for low spontaneous firing rates? Or equivalently, asymmetric inhibitory inputs to pairs and inhibitory input to trio for high spontaneous rates? It's not clear to me that the space spanned by (θ_{12} , θ_{123}) always provides sufficient resolution to distinguish network microarchitecture.

Thank you so much for this comment. We agree with all of the above comments. In order to distinguish architecture by ruling out the others, neurons must show significant interactions placed in the clearly separated area. As such, we started this discussion with "Our results point to the *possibility* of revealing".

Fig. 13 This is all interpretative and does not belong in the caption.

Removed.

L551 "among" should be italicized to emphasize that we are looking at connections between recorded neurons rather than inferring common inputs.

'among' was italicized.

L562 What does this mean exactly? You mean the possible variations in structure that the GLM framework is capable of discovering are limited?

We revised this sentence as follows.

"these statistical models are not directly constrained by physiological limits such as the Dales' law for the hidden common inputs, or physiological membrane dynamics and spiking thresholds that the LIF neuron model can offer~\cite[{}]{ladenbauer2019inferring}." "

L573 <word choice>

We replaced 'devices' with 'functions'.

L582 ...small (right) to reliably distinguish between input architectures.

We added 'to reliably distinguish between input architectures'. We also brought this paragraph to the Result section.

L620 Perhaps, but how would this then affect your conclusions in Figs. 5A and 5B?

If we have more than one signaling input in a bin, based on theoretical calculation (Eq. 37 in Shomali et. al, J CNS 2018), this effect is roughly the same as modulation of the amplitude of signaling input and shifts the amplitude to stronger values. It also can induce other effects such as decreasing the spontaneous spiking probability and increasing the spiking probability after signaling inputs. These effects are discussed in Supporting Information I, Fig. S1 and S2 (in the new version). We simulated LIF neurons receiving Poissonian common input (5 Hz) on top of noisy inputs in different motifs where the event of having two signaling inputs within a bin should happen. Although multiple signaling inputs in one bin is one cause of the deviation among others, we confirmed that the theory predicted the general trend of the interactions obtained by the simulation using the 5-10 ms time bin (Fig. S2). So it is not a practical problem in our study and the conclusion is not affected.

L677 Please expand on this! I see this issue (establishing ground-truth) as fundamental to the success of this framework, and its adoption by the community.

Thank you so much for requesting this. Since we agree with the reviewer that this is a fundamental point, we extended this sentence and put it as a separate paragraph as follows.

"Finally, we hope that the additional experimental evidence of the structured interactions (Fig.~\ref{fig:allen}) and theoretical predictions presented here motivate neurophysiologists to perform experiments that can directly identify input types and the network's structure in living animals. More specifically, although it is quite challenging, an experiment simultaneously performing in vivo patch-clamp of postsynaptic neurons and common inputs (while specifying the types of neurons, e.g., by genetic methods) could provide the ground truth of the architecture and is critical to improving the prediction of the proposed method."

REVIEWERS' COMMENTS:

Reviewer #1 (Remarks to the Author):

Thank you for the additions, very good.

1)

Thank you for adding the NEUORN multicompartiment simulations, it is a great addition. I would suggest also adding more information about it, I think it is lacking a bit more complete method. Specifically, where are the synapses that you are activating? E.g. what are the locations of the synapses on the dendrite and does it affect it? What was the synaptic weight (conductance) etc..., something like Table 1.

I would also strongly recommend publishing this simulation code. It will allow users to see an example usage of your tool.

9)

Thanks for that, very beneficial.

Why did you do it only on the multicompartimental model and not on the experimental/LIF data?

Anyway, seems that 1 minute is enough, is that a correct statement? If it is it should be mentioned in the Discussion or something of that kind.

THank you

Reviewer #2 (Remarks to the Author):

Summary:

The authors present an analytical framework for inferring local input architecture to neural trios based on higher-order (pair- and triple-wise) interaction terms in an information geometric measure applied to spiking statistics. They corroborate their analytical framework with an in silico of study of interaction strengths using detailed, multicompartimental models of L5 PC morphologies.

The authors use this framework to predict the most likely local connectivity scheme giving rise to measured spiking activity in anesthetized primate (macaque) V1, and awake recordings in mouse V1. They find that the strong negative triple-wise (but positive pair-wise) interactions observed in the original recordings are best explained by excitatory inputs to pairs of neurons in each trio, rather than by mutually shared inhibition (the intuitive explanation).

Furthermore, they probe the robustness of their results with respect to various assumptions including: 1) the model of neural spiking underlying their analytical framework, 2) the activity balance regime (near-threshold, subthreshold, superthreshold), 3) the presence of reciprocal/directional connections between neurons within the trio. In all cases, they find that interaction strengths are bounded by input amplitude on one side and postsynaptic spontaneous activity rates on the other. Finally, they compare the ability of their framework to distinguish among possible input architectures against more conventional methods (including pair-wise cross-correlation and pair-wise covariance), finding that their proposed framework consistently outperforms the alternatives.

Based on their analysis, they provide an interaction strength "guide map" to researchers who wish to replicate their approach. What follows below are general comments regarding content and

language/style. Please refer to the annotated pdf attached for additional detailed feedback.

General Comments (Content):

The immense care and effort on the part of the authors in preparing and undertaking this research is apparent. Robust inference of the local connectivity architecture in neuron trios based purely on spiking statistics (without relying on strong assumptions regarding neural models and/or proximity to threshold) is impressive and useful. Also, the textual revisions and additional simulations/analyses introduced between the previous and current versions of the manuscript are notable. In particular:

- 1) I had concerns about applicability in awake data. In response, the authors performed additional data analysis using the Allen Institute's visual coding datasets (awake mouse V1) to demonstrate the preponderance of positive pair-wise and negative triple-wise interactions, which are best explained by excitatory-to-pairs motif. Furthermore, they corrected for non-stationary firing rates using a Bayesian filtering and smoothing technique. This concern was therefore suitably addressed.
- 2) I was not convinced that uncertainty in the data had been sufficiently characterized to make claims about which theoretical boundaries best captured/explained the observed experimental interaction strength data. To address this concern, the authors quantified credible intervals for estimated interactions in the Allen Institute data to show that these intervals generally fell within the excitatory-to-pairs portion of the phase-plane.
- 3) The authors made significant structural alterations to the paper, moving various items to the SI, and standardizing language/notation. These changes improve the layout and readability of the manuscript.
- 4) Additional simulations of detailed Blue Brain L5 PC morphologies to verify the analytical predictions add credence to the theoretical aspects of the paper.

Several minor comments/criticisms can be found in the annotated pdf.

General Comments (Language/Style):

I can see that the word choice, grammar and notational consistency have been improved since the previous version. However, even with the improvements I have concerns that the language will distract readers from the content of your message. Many issues remain regarding number/case agreement, article use, word choice, verb tense/mood, and occasionally spelling and formatting. In your rebuttal, you mention that the manuscript has not yet been sent out for English proofreading, but that it will be prior to publication. Please give yourself ample time for this process, as I believe it will be an important influencing factor in the reception of your work.

Reviewers' comments: black

Our responses: blue

REVIEWERS' COMMENTS:

Reviewer #1 (Remarks to the Author):

Thank you for the additions, very good.

1) Thank you for adding the NEUORN multicompartment simulations, it is a great addition. I would suggest also adding more information about it, I think it is lacking a bit more complete method. Specifically, where are the synapses that you are activating? E.g. what are the locations of the synapses on the dendrite and does it affect it? What was the synaptic weight (conductance) etc..., something like Table 1.

I would also strongly recommend publishing this simulation code. It will allow users to see an example usage of your tool.

In Supplementary Note 5, we added further information on the parameters we used and the location of inputs and synapses on basal and apical branches of dendrites (Fig.S7 and Table.1 in the revised manuscript). We activated all of the synapses in the morphological model independently with a rate of 10 Hz. We selected one site (0.7 section of soma) for the shared signaling input stimulation (5 Hz rate). The stimulation pulse for common inputs is the exponential synapse with a decaying time constant 0.1 ms, and we changed the weight of shared signaling input from 0.1 to 4 (Fig. 6b, different colors). For all other synapses, the weight, time constant, synaptic types, activation sites, and other information are available in the file "synapses.tsv" provided by the blue brain project. Since each simulation needs considerable computational cost and time, we consider investigating the effects of varying the synaptic locations of shared and signaling inputs as a topic for future research.

We will also include the code for this multicompartment simulation by NEURON simulator with other codes for the analysis.

9) Thanks for that, very beneficial.

Why did you do it only on the multicompartmental model and not on the experimental/LIF data? Anyway, seems that 1 minute is enough, is that a correct statement? If it is it should be mentioned in the Discussion or something of that kind.

The duration required to obtain the desired accuracy in estimating neuronal interactions depends on the specific neuron models and synaptic inputs. Even for the multi-compartmental model that may give a realistic estimate of the required duration, the estimate is affected by the

synaptic noise level or amplitude of the shared inputs. Thus we can not conclude if 1 minute is enough in general for the morphological model or empirical data. We added the limitation of the morphological simulation in the explanation of Fig. S8.

What we think is more important is to present the uncertainty of the estimation for the given duration of the data at hand. For the experimental and LIF data, we gave the confidence bound/credible interval for the estimate of interactions (Fig.2B, 4AB; Fig.8).

Reviewer #2 (Remarks to the Author):

Summary:

The authors present an analytical framework for inferring local input architecture to neural trios based on higher-order (pair- and triple-wise) interaction terms in an information geometric measure applied to spiking statistics. They corroborate their analytical framework with an in silico study of interaction strengths using detailed, multicompartmental models of L5 PC morphologies.

The authors use this framework to predict the most likely local connectivity scheme giving rise to measured spiking activity in anesthetized primate (macaque) V1, and awake recordings in mouse V1. They find that the strong negative triple-wise (but positive pair-wise) interactions observed in the original recordings are best explained by excitatory inputs to pairs of neurons in each trio, rather than by mutually shared inhibition (the intuitive explanation).

Furthermore, they probe the robustness of their results with respect to various assumptions including: 1) the model of neural spiking underlying their analytical framework, 2) the activity balance regime (near-threshold, subthreshold, superthreshold), 3) the presence of reciprocal/directional connections between neurons within the trio. In all cases, they find that interaction strengths are bounded by input amplitude on one side and postsynaptic spontaneous activity rates on the other. Finally, they compare the ability of their framework to distinguish among possible input architectures against more conventional methods (including pair-wise cross-correlation and pair-wise covariance), finding that their proposed framework consistently outperforms the alternatives.

Based on their analysis, they provide an interaction strength "guide map" to researchers who wish to replicate their approach. What follows below are general comments regarding content and language/style. Please refer to the annotated pdf attached for additional detailed feedback.

General Comments (Content):

The immense care and effort on the part of the authors in preparing and undertaking this research is apparent. Robust inference of the local connectivity architecture in neuron trios

based purely on spiking statistics (without relying on strong assumptions regarding neural models and/or proximity to threshold) is impressive and useful. Also, the textual revisions and additional simulations/analyses introduced between the previous and current versions of the manuscript are notable. In particular:

1) I had concerns about applicability in awake data. In response, the authors performed additional data analysis using the Allen Institute's visual coding datasets (awake mouse V1) to demonstrate the preponderance of positive pair-wise and negative triple-wise interactions, which are best explained by excitatory-to-pairs motif. Furthermore, they corrected for non-stationary firing rates using a Bayesian filtering and smoothing technique. This concern was therefore suitably addressed.

2) I was not convinced that uncertainty in the data had been sufficiently characterized to make claims about which theoretical boundaries best captured/explained the observed experimental interaction strength data. To address this concern, the authors quantified credible intervals for estimated interactions in the Allen Institute data to show that these intervals generally fell within the excitatory-to-pairs portion of the phase-plane.

3) The authors made significant structural alterations to the paper, moving various items to the SI, and standardizing language/notation. These changes improve the layout and readability of the manuscript.

4) Additional simulations of detailed Blue Brain L5 PC morphologies to verify the analytical predictions add credence to the theoretical aspects of the paper.

Several minor comments/criticisms can be found in the annotated pdf.

Our replies to the minor comments in the PDF are attached below.

General Comments (Language/Style):

I can see that the word choice, grammar and notational consistency have been improved since the previous version. However, even with the improvements I have concerns that the language will distract readers from the content of your message. Many issues remain regarding number/case agreement, article use, word choice, verb tense/mood, and occasionally spelling

and formatting. In your rebuttal, you mention that the manuscript has not yet been sent out for English proofreading, but that it will be prior to publication. Please give yourself ample time for this process, as I believe it will be an important influencing factor in the reception of your work.

We sent the manuscript to a professional English proofreader and included the corrections in the revised manuscript attached to this response. In addition, we included the corrections you indicated in the PDF.

Response to minor comments in PDF

L86 Is it really unambiguous?

We agree that we needed the condition for this statement. For the common inputs with small amplitude, it is undoubtedly challenging to discern the underlying architecture because we can not distinguish the estimated small interactions from the sampling noise due to the finite data length. However, if the observed interactions are significant, we can unambiguously specify their basic architecture (excitatory-to-trio or excitatory-to-pairs) and reject the inhibition-to-trio. We thus revised this sentence as follows:

"Moreover, we provide model-free boundaries that each architecture occupies in the space of the neuronal interactions, with which one can unambiguously distinguish the underlying motif if the interactions are significant."

L96 The correct word is "plane", but I recommend "phase diagram" or something similar.

The 'phase diagram' is often used to describe the system showing the distinct statistical phases, often with phase transitions (in statistical physics). The neuronal activity here does not show such multiple phases, although we believe the underlying architecture is distinct. We also checked the 'phase plane.' However, it is used for the vector field of dynamical systems. We, therefore, wish to keep using the 'plane' of pairwise and triple-wise interactions.

L101 missing a space

added

L158 How much does this assumption affect the results?

In conclusion, we expect that this does not affect the boundaries we draw. We assumed that the onsets of common input are aligned at the bins in the simulation in Fig.2 for comparison with analytical results. When we assume the signal is aligned at the bins, the probability of having one or more spikes in a bin is calculated by integrating the first spike density from 0 to Δ , using the spiking density after the signaling input (Fig.1b red). Suppose the signal arrival is not aligned at the bins and happens in the middle of the bin. In that case, we need to use the first spiking densities without and with the signaling input before and after the signal arrival,

respectively. The probability of having a spike increases only after the excitatory signaling inputs. So, having the period without the signaling input in a bin due to an unaligned signaling input decreases the probability of having a spike within the bin compared to the aligned input. For inhibitory signaling input, the spiking probability after inhibitory signal decreases. So if the signal occurs in the middle of the bin, the first spiking density for that bin increases compared with when the signal occurs at the start of the bin. In both cases, as a consequence, the interactions are expected to be weaker than in the aligned cases. Changing the spiking probability can be effectively realized by changing the amplitude of the signaling input. Since we draw the boundary lines varying the amplitudes, we expect that this effect does not affect the conclusion of the manuscript.

L177 Equation 2 gives the probability of both neurons spiking within Δ for an input rate of λ (including the scenario where the neurons receive no input). But the notation is odd because in Equation 1 $P_A(x_1, x_2)$ is the probability of any sequence (including no spike), whereas in Equation 2, $P_A(x_1, x_2)$ really means $P_A(x_1=1, x_2=1)$, correct?

Thank you very much for pointing this out. We had an incorrect equation in the second line of Eq.2. We removed this equation. $P_A(x_1, x_2)$ in Eq.2 also gives the probability of generating all possible patterns specified by x_1 and x_2 (a probability mass function) as in Eq.1.

It gives the pattern probabilities in the bin right after the signaling input (note: neurons do not necessarily spike after the signaling input). The same is true for $P_0(x_1, x_2)$, the pattern probabilities where the signaling input does not exist. The pattern probability of the two neurons $P(x_1, x_2)$ (the left side of Eq.2) is given as the mixture of these two distributions with an appropriate proportion of such bins.

Fig.1 Broken link.

We will make sure the link works at the production stage.

L187 missing the word "where"

Inserted.

Fig. 2 Almost all the equation links appear to be broken. Please check all within-document hyperlinks.

We will make sure the link works at the production stage.

L224 This intuitive interpretation of the diffusion coefficient D would be nice to introduce earlier - i.e., the first time it is mentioned.

We revised the sentence in which D appears the first time (L130) as follows to make it clear that the diffusion coefficient D is related to the noise level of the background synaptic inputs.

"The noisy background inputs that the neuron receives are approximated by a Gaussian distribution with a mean drive of \bar{I} and the variance of $2D/\tau_m$, where D , τ_m , $[mV]^2$ is the diffusion coefficient. "

L288 This wording is wrong. Should be: "...increases $P(1,1,0)$ (and its permutations) in the denominator of Eq. 17, resulting in a negative value of θ_{123} ."

As written, it seems that increasing $P(1,1,0)$ should make θ_{123} LESS negative by attenuating the fraction that drives negativity (which is incorrect).

Corrected as suggested.

L306 on

Changed.

L350 How can we be sure the results weren't cherry-picked? I.e., many morphologies simulated and only the best selected? Could you include several different versions of 6B in the SI?

As a prototypical example, we choose the L5 neuron. We used the first pyramidal data of L5 in the website (<https://bbp.epfl.ch/nmc-portal>, L5:TTPC1:cADpyr232:1). We found it has a reasonable firing rate of ~20 Hz above the low-diffusion boundary we used in the manuscript (i.e., 1Hz spontaneous activity). This simulation requires significant computational resources. For the specific neuron that we chose (L5:TTPC1:cADpyr232:1), it took 3 months on our CPUs to complete to generate the whole data points, in addition to the learning period for the NEURON simulator and the Blue brain project code. We thus consider investigating different types of morphological architecture as the next step to elaborate our method.

L352 How did you choose synapse locations, and how many synapses per connection did you use? See comment on line 1453. (L1453 Where did you introduce them? How did you choose the locations?)

In Supplementary Note 5, we added further information on the synapses and the location of the signaling input (Supplementary Fig.7 and Supplementary Table 1 in the revised version). Please also see the response to reviewer #1's first question.

L356 An example of why it's perhaps unfair to claim on line 86 that your procedure unambiguously distinguishes among neural motifs.

As discussed in the answer to L86, it is challenging to discern the architecture for the common inputs with small amplitudes because we can not distinguish the estimated small interactions from the measurement noise. However, if the observed interactions are significant, we can unambiguously specify their basic architecture (excitatory-to-trio or excitatory-to-pairs) and reject the inhibition-to-trio. As mentioned above, we added the condition of the strong interactions to the sentence with 'unambiguously.'

L364 capitalization

Corrected

L463 Just to be clear - you do not claim you can distinguish asymmetric and symmetric excitatory-to-pairs motifs, correct? Perhaps it would be good to emphasize this point.

We remark that there are distinct regions for asymmetric and symmetric excitatory-to-pairs motifs under the homogeneous assumption. However, these regions are close. Thus, heterogeneity in connections may induce overlap. Therefore we do not claim in the manuscript that we can distinguish asymmetric and symmetric excitatory-to-pairs motifs.

To ensure that the excitatory-to-pairs motif includes both symmetric and asymmetric types and are considered as a single class, we added "(symmetric and asymmetric)" to this sentence.

"Figure~\ref{comparewithData} shows that empirical data of most nearby neurons (filled red symbols) coincide with regions associated with the motif of excitatory-to-pairs (symmetric and asymmetric)."

L544 Another example of how the procedure is not always unambiguous.

We agree that the regions are ambiguous if the probability of spiking is 0.5. In this regime, the interactions are small (Fig. S3) for both excitatory and inhibitory-based motifs, which makes the differentiation challenging. This situation is similar to the weak signaling input, where we observed small interactions from which we can not make a clear conclusion.

In the revised manuscript, we noted that "one can unambiguously distinguish the underlying motif if the interactions are significant." This remark on the condition applies to this case, too.

L576 Why the italics?

Removed

L585 italics?

Removed

L586 italics?

Removed

L623 spelling (guide)

Corrected

L635 Is this mentioned somewhere in the Results? If not, please include.

We added our analysis on the mixing motifs in the Result section.

L719 italics?

Removed

L720 should be italicized

Removed